# Direct activation of HSF1 by macromolecular crowding and misfolded proteins

Oliver Simoncik[1,2], Vlastimil Tichy[1], Michal Durech[1¤a], Lenka Hernychova[1], Filip Trcka[1¤b], Lukas Uhrik[1], Miroslav Bardelcik[1,3], Philip J. Coates[1], Borivoj Vojtesek[1], Petr Muller[1] *

**1** Research Centre for Applied Molecular Oncology (RECAMO), Masaryk Memorial Cancer Institute, Brno, Czech Republic, **2** Department of Experimental Biology, Faculty of Science, Masaryk University, Brno, Czech Republic, **3** Department of Biochemistry, Faculty of Science, Masaryk University, Brno, Czech Republic

¤a Current address: Department of Hematooncology, University of Ostrava, Ostrava, Czech Republic
¤b Current address: Institute of Microbiology, Czech Academy of Sciences, Prague, Czech Republic
* muller@mou.cz

**Data Availability Statement:** The mass spectrometry proteomics data have been deposited to the ProteomeXchange Consortium via the PRIDE (proteomics identifications database)

## Abstract

Stress responses play a vital role in cellular survival against environmental challenges, often exploited by cancer cells to proliferate, counteract genomic instability, and resist therapeutic stress. Heat shock factor protein 1 (HSF1), a central transcription factor in stress response pathways, exhibits markedly elevated activity in cancer. Despite extensive research into the transcriptional role of HSF1, the mechanisms underlying its activation remain elusive. Upon exposure to conditions that induce protein damage, monomeric HSF1 undergoes rapid conformational changes and assembles into trimers, a key step for DNA binding and transactivation of target genes. This study investigates the role of HSF1 as a sensor of proteotoxic stress conditions. Our findings reveal that purified HSF1 maintains a stable monomeric conformation independent of molecular chaperones in vitro. Moreover, while it is known that heat stress triggers HSF1 trimerization, a notable increase in trimerization and DNA binding was observed in the presence of protein-based crowders. Conditions inducing protein misfolding and increased protein crowding in cells directly trigger HSF1 trimerization. In contrast, proteosynthesis inhibition, by reducing denatured proteins in the cell, prevents HSF1 activation. Surprisingly, HSF1 remains activated under proteotoxic stress conditions even when bound to Hsp70 and Hsp90. This finding suggests that the negative feedback regulation between HSF1 and chaperones is not directly driven by their interaction but is realized indirectly through chaperone-mediated restoration of cytoplasmic proteostasis. In summary, our study suggests that HSF1 serves as a molecular crowding sensor, trimerizing to initiate protective responses that enhance chaperone activities to restore homeostasis.

## Introduction

Heat shock factor 1 (HSF1) is a stress-response transcription factor that plays a crucial role in maintaining proteostasis within cells by responding to various stressors that induce protein

repository (44) with the dataset identifier "PXD037662"; Project Webpage: http://www.ebi.ac.uk/pride/archive/projects/PXD037662 FTP Download: https://ftp.pride.ebi.ac.uk/pride/data/archive/2024/08/PXD037662 Further information and requests for resources and reagents should be directed to and will be fulfilled by the Lead Contact, P. Muller (muller@mou.cz).

**Funding:** The project was supported by the project National Institute for Cancer Research (Programme EXCELES, ID Project No. LX22NPO5102)—funded by the European Union—Next Generation EU and by Ministry of Health Development of Research Organisation, MH CZ - DRO (MMCI, 00209805). O. S., M.B. and P.M. were supported by the Czech Science Foundation (22-17102S), B.V. was supported by the Czech Science Foundation (22-02940S). The funders had no role in study design, data collection and analysis, decision to publish, or preparation of the manuscript. There was no additional external funding received for this study.

**Competing interests:** The authors have declared that no competing interests exist.

denaturation and aggregation. Activation of HSF1 occurs in response to various physical and chemical stresses, as well as specific inhibitors targeting the proteasome or chaperones. It acts as a transcription factor orchestrating the expression of stress-related proteins, including chaperones like Hsp70 and Hsp90, vital for cellular proteostasis and considered potential anti-tumor therapeutic targets [1].

HSF1's involvement in malignant transformation underscores its crucial role in mitigating stresses associated with increased protein synthesis and genomic instability in cancer cells [2, 3]. Despite extensive research, key aspects governing HSF1 activation remain enigmatic. The formation of trimeric HSF1, crucial for its DNA-binding and transcriptional activity, is triggered by various stressors, yet the mechanisms underlying activation are not fully elucidated.

Three scenarios have been proposed to explain HSF1 activation, each with their own set of evidence and unanswered questions. While it has been suggested that HSF1 acts as a stand-alone stress sensor in response to elevated temperatures [4, 5], its direct activation by other stress types remains unclear (Fig 1A). Alternatively, HSF1 may function as a central node that integrates multiple stress signaling pathways, although the identity of other proteotoxic stress sensors remains elusive [3]. The role of posttranslational modifications in HSF1 activation is debated, and it is unclear if they initiate or result from trimerization. Multiple phosphorylation site mutations in HSF1 did not prevent activation during heat shock, and a genome-wide RNAi screen found no evidence for kinase regulation [6, 7]. Thus, despite being a hallmark of the heat shock response, the precise role of HSF1 phosphorylation remains elusive, suggesting it may fine-tune rather than trigger trimerization (Fig 1B). The third possibility posits that HSF1 activation is controlled by interactions with chaperones, particularly Hsp90 and Hsp70, which regulate HSF1 trimerization and subsequent transcriptional activity. Cellular stress displaces chaperones, allowing HSF1 to form active trimers. This hypothesis suggests that HSF1 remains inactive in a monomeric state bound to chaperones, which are disrupted by Hsp90 inhibitors [8]. However, recombinant HSF1 can remain monomeric independent of chaperones [4], and Hsp90 does not significantly affect its activation temperature (Fig 1C). Rather than being mutually exclusive, these scenarios might complement each other, and our research is focused on uncovering how they integrate to regulate HSF1 activation.

Our study aims to shed light on these mechanisms by elucidating how cells recognize disturbances in proteostasis and identifying universal mechanisms essential for HSF1 activation. Through a series of in vitro and in vivo experiments, we investigate the role of various stress conditions in HSF1 activation and explore the underlying molecular mechanisms driving its conformational changes and trimerization.

Our investigation unveils a new aspect of HSF1 activation: its sensitivity to macromolecular crowding. By subjecting HSF1 to a protein-based crowding environment, we observe a marked increase in trimerization and DNA binding, both in vitro and in vivo. Our interaction studies reveal no significant decrease in chaperone-HSF1 binding under proteotoxic stress conditions, challenging theories positing chaperone dissociation as a prerequisite for HSF1 trimerization.

In elucidating the role of macromolecular crowding as a key determinant of HSF1 trimerization, our study advances the understanding of cellular stress responses and offers insights into potential therapeutic strategies targeting stress-responsive pathways in disease states.

## Materials and methods

### DNA constructs

The coding sequences of human HSF1 (NM_005526.4), Hsp70 (NM_005345.6), and Hsp90$\alpha$ (NM_005348) were cloned into pDONR221 using Gateway cloning technology (Thermo Fisher Scientific Inc., Waltham, MA, USA). The resulting plasmids were further cloned into expression

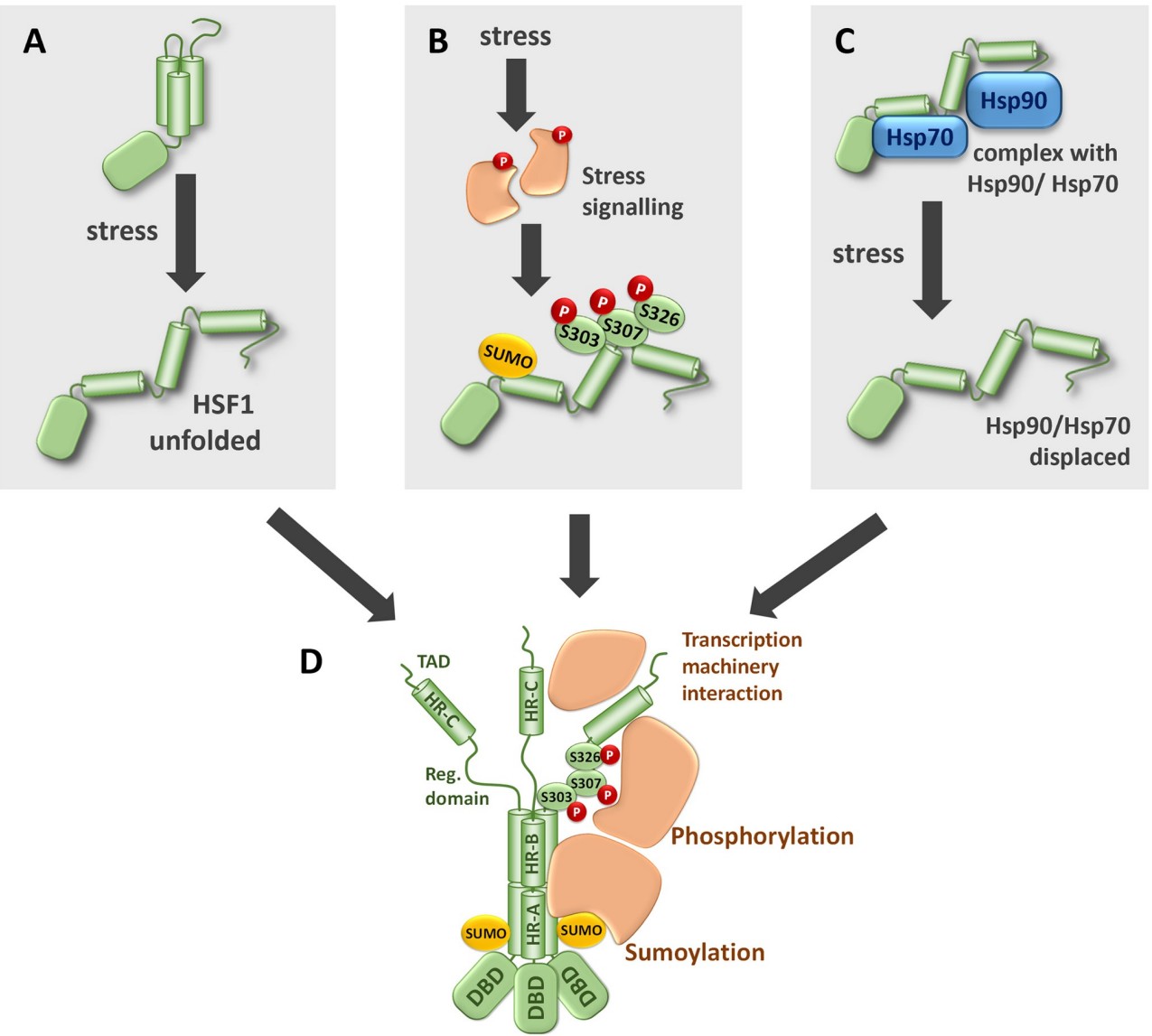

**Fig 1. Models of HSF1 activation.** A: HSF1 works as a direct sensor of stress, where diverse stress conditions lead to unfolding of the monomeric inactive conformation and stabilization of active trimers. B: Stress signaling pathways induce post-translational modifications that result in HSF1 activation. C: HSF1—Hsp70/Hsp90 complexes are disrupted in response to stress. D: Activated HSF1 forms trimers that bind DNA and recruit proteins responsible for transcription initiation, splicing and/or chromatin remodeling.

plasmids via Gateway LR reaction. pEXP17-His6-TEV-HSF1 was used to express HSF1 in bacteria, while pLenti-N-mCherry-HSF1, pLenti-N-SBP-Hsp70-IRES-GFP, and pLenti-N-SBP-Hsp90-IRES-GFP were used to produce lentiviral vectors. Lentivirus was prepared in 293FT cells using ViraPower™ Lentiviral Packaging Mix (Thermo Fisher Scientific Inc.). Annotated sequences of the plasmids are included in supporting information (S1–S6 Files).

## Cell lines cultivation and treatment

The human non-small cell lung carcinoma (NSCLC) cell line H1299 (ATCC CRL-5803) and the human melanoma cell line A375 (ATCC CRL-1619) were obtained from the American

Type Culture Collection (ATCC), Manassas, VA 20110, USA. Cells were cultured in Dulbecco's modified Eagle's medium with 10% fetal bovine serum, 300 mg/l L-glutamine, penicillin/streptomycin, and were grown to 80% confluence before experimental treatments. Stable cell lines expressing HSF1 fused to mCherry, SBP-Hsp70, and SBP-Hsp90 were prepared by transduction of lentiviral vectors using spinfection [9]. Cells were selected with puromycin (2 μg/ml) 2 days after transfection. Clones with stable expression of mCherry-HSF1 were selected using a BD FACSAria III cell sorter.

Knockout of HSF1 and HSF2 was performed by electroporation of synthetic sgRNA and recombinant Cas9 protein into H1299 cells. $5 \times 10^5$ cells were mixed with 8 μg Cas9 protein and 1.6 μg sgRNA in 100 μl electroporation buffer (4 mM KCl, 10 mM $MgCl_2$, 120 mM $NaH_2PO_4/Na_2HPO_4$, 10 mM HEPES pH 7.2). The cells were electroporated using an Amaxa electroporator, program X-005. HSF1 knockout was targeted by synthetic sgRNA sequence AGCTTCCACGTGTTCGACCA (Chr.8: 144308915—144308937 on GRCh38), and HSF2 by sequence GAAGCAGAGTTCGAACGTGC (Chr.6: 122399740—122399762 on GRCh38). Gene knockout was confirmed by sequencing of both alleles.

## Protein expression and purification

Human HSF1 cDNA was cloned into the pDEST17 vector containing an N-terminal $His_6$ tag, cleavable by TEV protease. The vector was transformed into BL21 (DE3) RIPL competent cells for protein production. Transformed cells were cultured in Luria-Bertani (LB) medium at 37°C with shaking at 140 RPM until $OD_{600}$ reached 0.3. The culture was cooled to 20°C for 40 minutes, and expression was induced with 0.25 mM IPTG for 3 hours. Cells were harvested by centrifugation at 6000 g for 15 minutes at 4°C.

The cell pellet was resuspended in chilled lysis buffer (25 mM HEPES pH 7.4, 0.3 M NaCl, 3 mM $\beta$-mercaptoethanol, 10% glycerol, 1 mg/ml lysozyme, 1 mM PMSF (phenylmethylsulfonyl fluoride), 1 mM $MgCl_2$, Turbonuclease (1:50000)) and lysed by sonication. The lysate was clarified by centrifugation at 12000 g for 30 minutes at 4°C.

$His_6$-TEV-HSF1 protein was purified using a 5 ml HisTrap FF column (Cytiva) on an ÄKTA pure L1 chromatography system (Cytiva). After washing with equilibration buffer (25 mM HEPES pH 7.4, 0.3 M NaCl, 3 mM $\beta$-mercaptoethanol, 10% glycerol), bound proteins were eluted with equilibration buffer containing 250 mM imidazole. The eluted protein was dialyzed overnight at 4°C against the equilibration buffer and subjected to TEV protease cleavage under the same conditions. To remove $His_6$-TEV protease, the sample was passed through a 1 ml HisTrap FF column (Cytiva), and the flow-through containing the cleaved HSF1 was collected. The protein was concentrated and buffer-exchanged to SEC buffer (25 mM HEPES pH 7.4, 0.3 M NaCl, 3 mM DTT, 10% glycerol) and purified by size exclusion chromatography using a Superdex 200 Increase 10/300 GL column (Cytiva). SEC was performed at 4°C with a flow rate of 0.3 ml/min. To avoid aggregation, purified HSF1 was not concentrated further, and fractions were flash-frozen and stored at -80°C.

## Fluorescence polarization binding assay

The equilibrium bindings of a fixed concentration (30 nM) of a fluorescent double-stranded DNA oligonucleotide 5'FAM-CCCCTTCCCGAATATTCCCCC-3' (FAM-HSE, fluorescein amidite-modified heat shock element) and increasing concentrations of HSF1 protein in buffer containing 150 mM NaCl, 50 mM HEPES pH 7.2, 1 mM dithiothreitol, and 0.05% Tween-20 were monitored by both total fluorescence intensity and fluorescence polarization on a Tecan Infinite M1000 plate reader (Tecan Group AG, Mannedorf, Switzerland) at 25°C. The equilibrium binding of a fixed concentration (30 nM) of fluorescent oligonucleotide and

increasing concentrations of either HSF1 monomer or trimer were monitored by fluorescence polarization. The molar concentration of the HSF1 trimer was calculated as one-third of the molar concentration of the HSF1 monomer at the same mass concentration.

The equilibrium dissociation constant ($K_d$) was then calculated by fitting the sigmoidal dose-dependent fluorescence polarization increases as a function of protein concentrations using GraphPad Prism and the following equation [10]:

$$Y = FP_0 + (FP_{max} - FP_0) \times \frac{(K_d + L_t + P_t) - \sqrt{(K_d + L_t + P_t)^2 - 4 \cdot L_t \cdot P_t}}{2 \cdot L_t} \tag{1}$$

where:

- $Y$ is the observed fluorescence polarization,—$FP_0$ is the fluorescence polarization of the free ligand,—$FP_{max}$ is the fluorescence polarization of the bound ligand,—$K_d$ is the dissociation constant,—$L_t$ is the total concentration of ligand (30 nM),—$P_t$ is the total concentration of protein (variable).

## Hydrogen/deuterium exchange (HDX) mass spectrometry

Monomeric, trimeric, and heat-treated HSF1 (42°C for 20 minutes) recombinant proteins (2 µM final concentration) were diluted into 25 mM HEPES pH 7.4, 150 mM NaCl, 10% glycerol, and 2 mM DTT in $H_2O$ to prepare undeuterated controls and for peptide mapping. Deuterium labeling was initiated by 5-fold dilution into 25 mM HEPES pH 7.0, 150 mM NaCl, 10% glycerol, and 2 mM DTT in $D_2O$. Hydrogen-deuterium exchange was carried out at 4°C and was quenched after 30, 600, and 3600 seconds by adding 0.875 M HCl in 1 M glycine (pH 2.3) and rapidly freezing in liquid nitrogen.

Each sample was thawed and injected into an LC system (UltiMate 3000 RSLCnano, Thermo Scientific) with an immobilized pepsin enzymatic column (Affipro s.r.o., CZ) with a bed volume of 15 µl and a flow rate of 100 µl/min in 2% acetonitrile/0.05% trifluoroacetic acid. Peptides were trapped and desalted on-line on a peptide microtrap (Michrom Bioresources, Auburn, CA) for 3 minutes at a flow rate of 100 µl/min. Next, the peptides were eluted onto an analytical column (Jupiter C18, 0.5 x 50 mm, 5 µm, 300 Å, Phenomenex, Torrance, CA) and separated by linear gradient elution starting with 10% buffer B in buffer A and rising to 40% buffer B over 17 minutes at a flow rate of 50 µl/min. Buffers A and B consisted of 0.1% formic acid in water and 80% acetonitrile/0.08% formic acid, respectively. The immobilized pepsin columns, trap cartridge, and analytical column were kept at 1°C.

Mass spectrometric analysis was carried out using an Orbitrap Elite mass spectrometer (Thermo Fisher Scientific) with electrospray ionization connected on-line to a robotic system based on the HTS-XT platform (CTC Analytics, Zwingen, Switzerland). The instrument was operated in a data-dependent mode for peptide mapping (HPLC-MS/MS). Each MS scan was followed by MS/MS scans of the three most intensive ions from both CID and HCD fragmentation spectra. Tandem mass spectra were searched using SequestHT against the cRAP protein database (ftp://ftp.thegpm.org/fasta/cRAP) containing the sequences of HSF1 (monomeric and trimeric), recombinant proteins with the following search settings: mass tolerance for precursor ions of 10 ppm, mass tolerance for fragment ions of 0.6 Da, no enzyme specificity, two maximum missed cleavage sites, and no fixed or variable modifications. The false discovery rate at the peptide identification level was set to 1%. Sequence coverage was analyzed with Proteome Discoverer version 1.4 (Thermo Fisher Scientific) and graphically visualized with the MS Tools application [11].

Analysis of deuterated samples was done in HPLC-LC-MS mode with ion detection in the orbital ion trap. The MS raw files, together with the list of peptides (peptide pool) identified with high confidence characterized by requested parameters (amino acid sequence of each peptide, its retention time, XCorr, and ion charge), were processed using HDExaminer version 2.2 (Sierra Analytics, Modesto, CA). The software analyzed protein and peptide behavior, created the uptake plots that showed peptide deuteration over time with calculated confidence levels (high and medium confidence are accepted, low confidence is rejected). The mass spectrometry proteomics data have been deposited to the ProteomeXchange Consortium via the PRIDE repository with the dataset identifier "PXDPXD037662"; Username: reviewer_px-d037662@ebi.ac.uk, Password: ChSpggcg. Additionally, we have uploaded the HDX data (S1 Fig), detailing peptide uptake for HSF1 states.

## High resolution clear native electrophoresis (HR CNE)

Oligomeric states of cellular mCherry-HSF1 were determined by an optimized protocol for high resolution clear native electrophoresis (HR CNE), originally developed by Wittig et al. [12]. Anode buffer (25 mM imidazole, pH 7.0) was placed in the outer chamber, while fresh cathode buffer (50 mM Tricine, pH 7.0, 7.5 mM imidazole, 0.05% sodium deoxycholate, 0.05% Triton X-100) was placed in the inner chamber of an electrophoretic apparatus. Cell pellets containing mCherry-HSF1 were resuspended in chilled lysis buffer (50 mM imidazole, pH 7.0, 150 mM NaCl, 2 mM $\beta$-mercaptoethanol, 2 mM MgCl$_2$, 1% Triton X-100, protease inhibitor cocktail (1:100), Turbonuclease (1:10000), and 1 mM PMSF) and lysed on ice for 30 minutes. Cell lysates were obtained by centrifugation in a pre-cooled centrifuge (4°C) for 10 minutes at 8000 g. Cell lysates were mixed with 2x loading buffer (700 µl cathode buffer, 300 µl glycerol, traces of ponceau S, and 2 mM EDTA) and separated on 4–10% polyacrylamide gradient gels, prepared as described by Wittig et al. [13]. The electrophoretic apparatus was held on ice, and separation was carried out at a constant voltage of 110 V for 2 hours. Native gels were subsequently scanned using a Typhoon FLA 9500 fluorescence imager.

## Blue native polyacrylamide gel electrophoresis (BN PAGE)

Oligomeric states of SEC fractions containing purified HSF1, as well as subsequent experiments using purified HSF1 and BSA, were determined by an optimized protocol for Blue native gel electrophoresis (BN-PAGE), originally developed by Wittig et al. [13]. Anode buffer (25 mM imidazole, pH 7.0) was placed in the outer chamber, while fresh cathode buffer (50 mM Tricine, pH 7.0, 7.5 mM imidazole, 0.002% Coomassie Brilliant Blue G-250) was placed in the inner chamber of an electrophoretic apparatus. For sample preparation, recombinant purified HSF1 (5 µM) was mixed with or without BSA (0–50 mg/ml) and kept on ice or incubated at elevated temperatures using a thermocycler. Following incubation, samples were immediately transferred to ice. The samples were diluted five-fold with buffer containing 25 mM HEPES, pH 7.4, 150 mM NaCl, 3 mM $\beta$-mercaptoethanol, and 10% glycerol. The diluted samples were mixed with 2x loading buffer (700 µl cathode buffer, 300 µl glycerol, 0.004 g Coomassie Brilliant Blue G-250, 2 mM EDTA) and separated on 7% polyacrylamide gels as described by Wittig et al. [13]. The electrophoretic apparatus was kept on ice, and separation was carried out at a constant voltage of 110 V for 2 hours. Native gels were subsequently used for immunoblotting.

## Electrophoretic Mobility Shift Assay (EMSA)

Purified HSF1 (200 nM) was mixed with Cy5-labeled HSE oligonucleotide (100 nM; 5'Cy5-CCCCTTCCCGAATATTCCCCC-3') in binding buffer (25 mM HEPES, pH 7.4, 150

mM NaCl, 10% Glycerol, 3 mM $\beta$-mercaptoethanol). Samples were incubated at various temperatures for different times according to experimental requirements. After incubation, samples were placed on ice for 30 minutes, diluted five-fold with binding buffer, and 5 μl of each sample was loaded onto a pre-chilled 1% agarose gel in TBE buffer. Electrophoresis was conducted at 135 V for 30 minutes at 4°C. The gel was subsequently imaged using a Typhoon FLA 9500 fluorescence imager.

To test DNA binding activity of SEC fractions, 2.5 μl of each fraction was premixed with 100 nM Cy5-HSE in binding buffer in a total volume of 40 μl. Samples were either kept on ice or incubated at 40°C for 10 minutes before being placed on ice for 30 minutes. Following incubation, samples were diluted five-fold with binding buffer, and 5 μl of each sample was loaded onto a pre-chilled 1% agarose gel in TBE buffer and processed as described above.

For time-dependent crowding experiments, 200 nM monomeric HSF1 (fraction C11) was premixed with 100 nM Cy5-HSE, with or without 50 mg/ml albumin. Samples were incubated at 39°C for varying times (0–600 seconds) and then placed on ice for 30 minutes. Samples were diluted five-fold with binding buffer, and 5 μl of each sample was loaded onto a pre-chilled 1% agarose gel. Electrophoresis and imaging were performed as described above.

For concentration-dependent crowding experiments, 200 nM monomeric HSF1 was premixed with 100 nM Cy5-HSE and increasing concentrations of albumin (0–100 mg/ml). Samples were incubated at 38°C for 5 minutes and then placed on ice for 30 minutes. Following incubation, samples were diluted five-fold with binding buffer, and 5 μl of each sample was loaded onto a pre-chilled 1% agarose gel. Electrophoresis and imaging were performed as described above.

## Nuclear/cytoplasmic fractionation

Cell pellets containing $2 \times 10^6$ cells were resuspended in 100 μl of cold Harvest buffer containing 10 mM HEPES (pH 7.9), 50 mM NaCl, 0.5 M sucrose, 0.1 mM EDTA, 0.5% Triton X-100, and freshly added 1 mM DTT, 1x protease inhibitor cocktail, and 1 mM PMSF. After 15 minutes at 4°C, nuclei were pelleted at 600 g in a swinging bucket rotor for 10 minutes. The supernatants containing cytoplasmic proteins were transferred to new tubes, and nuclear pellets were resuspended in 500 μl of buffer A containing 10 mM HEPES (pH 7.9), 10 mM KCl, 0.1 mM EDTA, 0.1 mM EGTA, and freshly added 1 mM DTT, 1x protease inhibitor cocktail, and 1 mM PMSF. After pelleting at 600 g in a swinging bucket rotor, the supernatants were discarded, and the pellets were resuspended in 100 μl of buffer C containing 10 mM HEPES (pH 7.9), 500 mM NaCl, 0.1 mM EDTA, 0.1 mM EGTA, 0.1% NP-40, and freshly added 1 mM DTT, 1x protease inhibitor cocktail, and 1 mM PMSF. After 15 minutes of vortexing at 4°C, samples were centrifuged at 14000 rpm for 10 minutes at 4°C. The supernatants containing nuclear proteins were transferred to new tubes. Cytoplasmic and nuclear extracts were analyzed by immunoblotting using anti-HSF1 mouse mAb c-5 (Santa Cruz Biotechnology Inc.). Fluorescence of mCherry-HSF1 was assessed in nuclear and cytoplasmic fractions at excitation 587/5 nm and emission 610/5 nm using a plate reader Infinity M1000Pro (Tecan Group Ltd., Männedorf, Switzerland).

## Real-time cell impedance

For each experiment, $3 \times 10^4$ H1299 cells in 100 μl DMEM were transferred per well to an E-Plate VIEW 16 (xCELLigence; Invitrogen, Santa Clara, CA, USA). The plate was placed in an RTCA analyzer in a humidified incubator, and impedance monitoring was started. Drugs were added after 8 hours; final concentrations were 200 nM for Luminespib (AUY-922), BIIB021, Ganetespib, Tanespimycin (17-AAG), Onalespib (AT13387), and Geldanamycin-

FITC, or 10 μg/ml for cycloheximide, all dissolved in DMEM. Impedance was measured for 12 hours.

## Pull-down assay

$1 \times 10^7$ H1299 cells stably expressing either SBP-tagged Hsp70 (HSPA1A) or SBP-tagged Hsp90 (HSP90AA1) were treated with 10 μg/ml cycloheximide for 30 minutes. Subsequently, cells were co-treated with either 300 nM AUY-922, 1 μM bortezomib, or heat shock at 42˚C for an additional 1 hour. Cells were washed with cold PBS, scraped into 5 ml cold PBS, and pelleted by centrifugation at 1000 g for 5 minutes at 4˚C. Cell pellets were lysed using wash buffer (20 mM HEPES pH 7.4, 150 mM potassium acetate, 2 mM $MgCl_2$) supplemented with 1% Triton X-100, 10 μg/ml avidin, protease inhibitor cocktail (1:100), and 1 mM PMSF, and were further processed through sonication. After centrifugation at 10000 g for 10 minutes at 4˚C, lysates were transferred to new tubes and incubated with 20 μl high-capacity streptavidin agarose resin beads (Thermo Scientific) for 2 hours at 4˚C. Prior to adding samples, beads were equilibrated with wash buffer in three wash steps, followed by centrifugation at 8000 g for 1 minute each time. After incubation with lysate, beads were washed three times with 1 ml of wash buffer, followed by centrifugation at 8000 g for 1 minute each time. Elution was performed using elution buffer, which consisted of wash buffer supplemented with 1 mM biotin. Elution lasted for 5 minutes at room temperature. Beads were spun down by centrifugation at 8000 g for 1 minute, and the resulting eluates were transferred to new tubes. To account for nonspecific protein binding to the matrix, a parallel experiment was conducted using streptavidin beads pre-blocked with 1 mM biotin. The levels of HSF1 copurified with SBP-tagged chaperones were assessed using western blotting and monoclonal rabbit anti-HSF1 antibody D3L8 (#12972, Cell Signaling, Danvers, MA, United States).

## Results

### Purified HSF1 maintains a compact monomeric conformation that spontaneously trimerizes and binds DNA in response to elevated temperatures

Our primary objective was to investigate the behavior of monomeric HSF1, particularly its stability and response to stress conditions, using purified recombinant HSF1. Through size exclusion chromatography followed by Blue native electrophoresis, we confirmed that HSF1 purified from bacteria maintains a stable monomeric conformation independently of chaperone stabilization under normal conditions (Fig 2A and 2B). Blue native electrophoresis further demonstrated that heat shock alone induces the transition of HSF1 from monomeric to trimeric state (Fig 4A). S2 Fig provides a more detailed characterization of the separated monomer and trimer fractions. We initially evaluated DNA binding capacity using fluorescence polarization, which revealed that the transition of HSF1 from monomers to trimers enhances DNA binding in response to heat stress (Fig 2E and 2F). To further substantiate these findings, we employed electrophoretic mobility shift assay (EMSA). This technique consistently showed that HSF1 acts as a primary sensor of heat stress, undergoing trimerization and subsequent DNA binding upon exposure to elevated temperatures (Fig 2C and 2D). These in vitro findings suggest that the monomeric HSF1 remains stable and functions as a primary sensor of heat stress, even in the absence of additional chaperones or co-factors.

Molecular structural alterations in HSF1 following heat stress were further analyzed using hydrogen-deuterium exchange (HDX) mass spectrometry. The results showed that the conformation of the thermally induced trimer is identical to that of the trimer isolated during

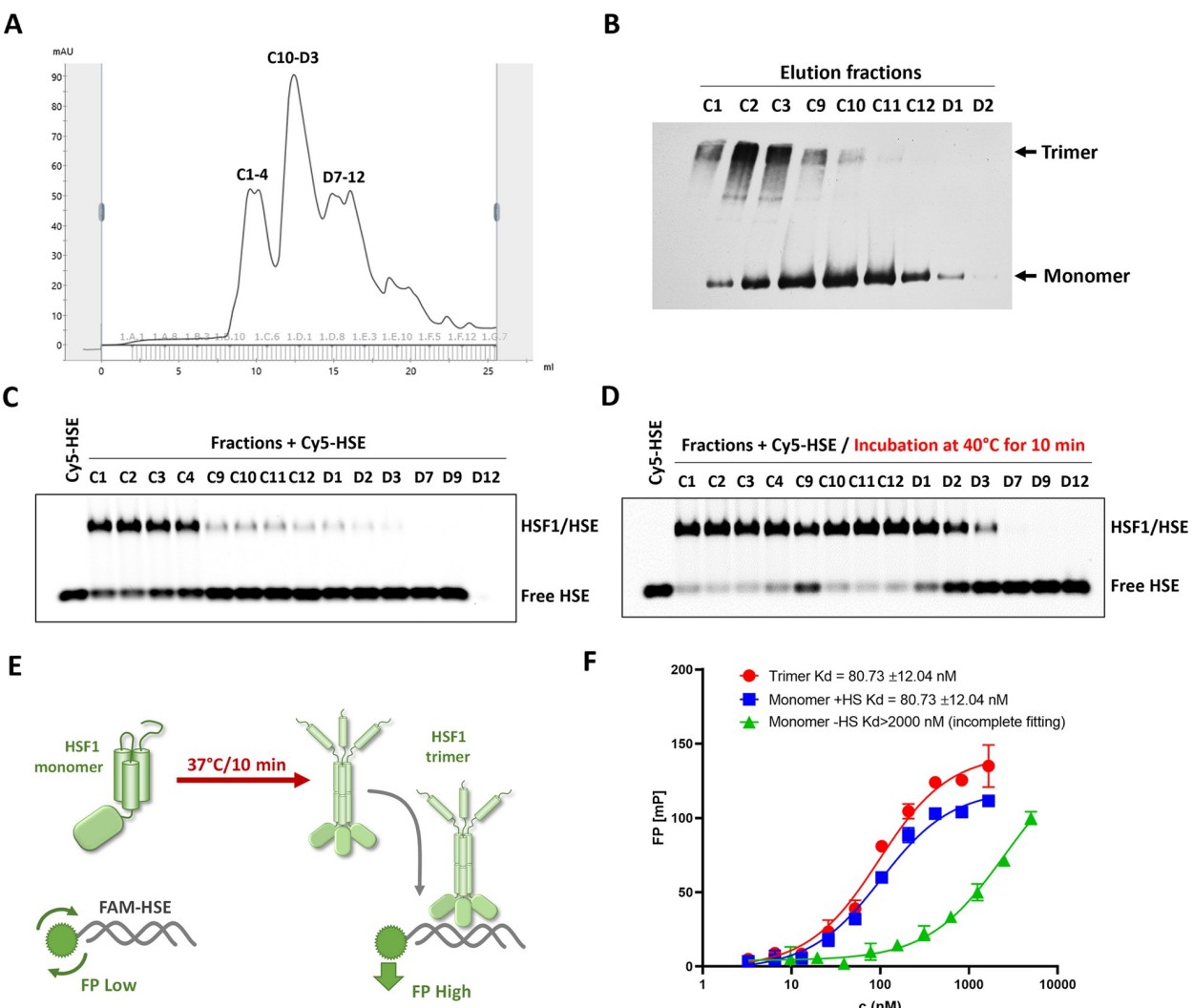

**Fig 2. Inactive HSF1 forms a compact monomeric conformation that is disrupted by stress leading to trimer assembly and DNA binding.** A: Monomeric HSF1 was separated from trimers using size exclusion chromatography (SEC). B: The oligomeric states of HSF1 in the SEC separated fractions were evaluated using Blue native electrophoresis followed by western blotting. C: EMSA showing the DNA binding activity of SEC fractions mixed with Cy5-labeled oligonucleotide containing a canonical heat shock element (HSE). D: EMSA of SEC fractions mixed with Cy5-labeled HSE oligonucleotide, with samples incubated at 40°C for 10 minutes prior to electrophoresis, demonstrating the effect of heat treatment on HSF1 DNA binding. E: Schematic version of fluorescence polarization (FP) assay to measure DNA binding of purified HSF1 with or without heat shock activation using FAM-labeled DNA containing a HSE sequence. F: DNA binding capacity of HSF1 monomers with (+HS) or without heat shock (-HS) was compared to that of HSF1 trimers by FP. FP is measured as ratio of polarized light intensities in units of mP.

bacterial purification, suggesting that chaperones and post-translational modifications are not required for trimer formation. The most significant differences between the trimer and monomer were observed in the HR-A and HR-B regions, where strong protection occurred due to oligomerization. In contrast, the HR-C region exhibited opposite changes, with relatively less deuteration in monomers and increased exposure to solvent in trimers (Fig 3). These findings suggest that HR-C contributes to stabilizing the monomeric conformation of HSF1, consistent with previous studies [4]. The results obtained from purified HSF1 indicate that trimerization and DNA binding activation rely primarily on stress-induced conformational changes without the requirement for additional proteins. Notably, stress-induced conformational changes occur independently of post-translational modifications.

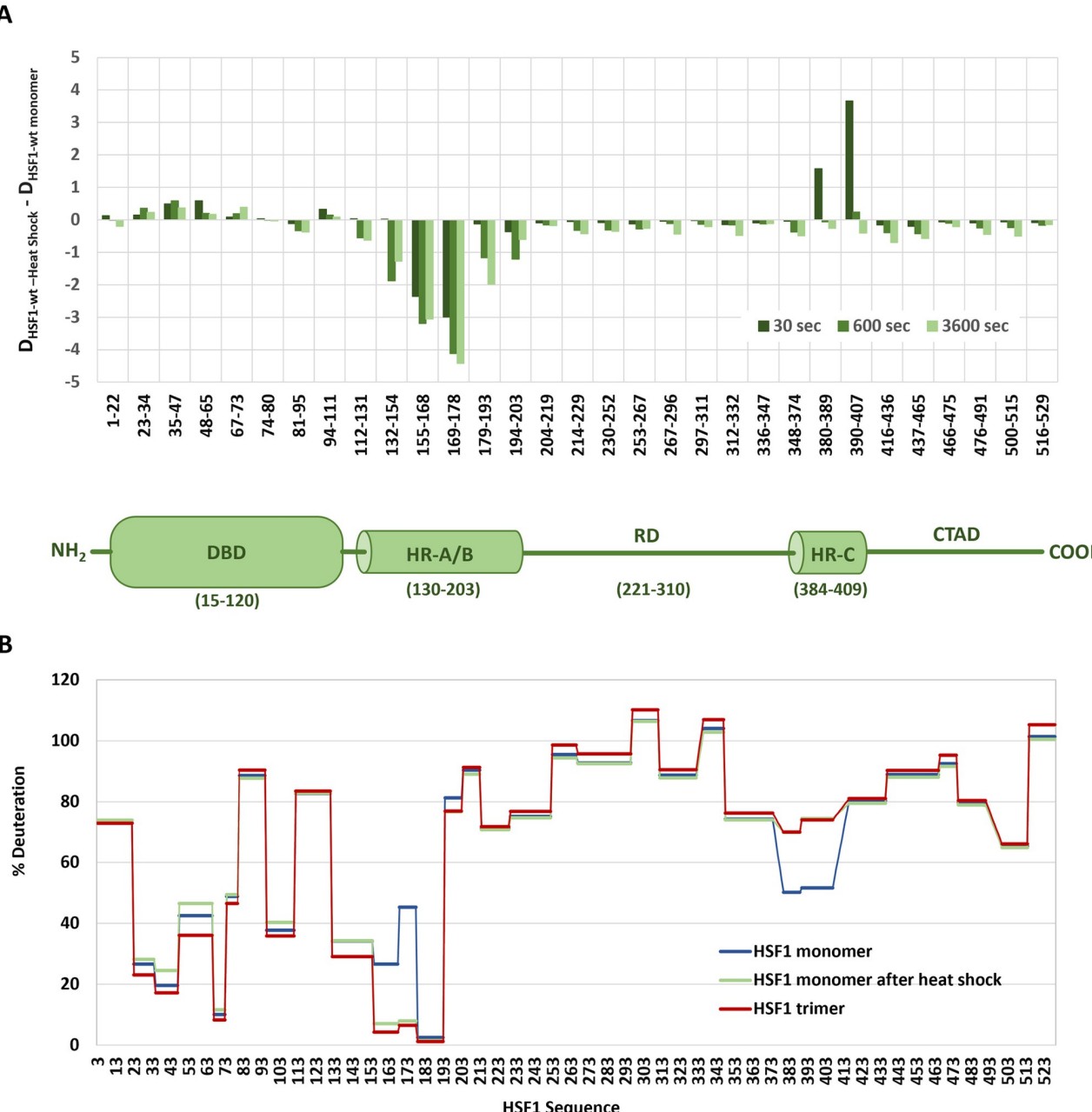

**Fig 3. Analysis of HSF1 conformational changes using HDX.** The HSF1 monomeric fraction, purified by SEC, was subjected to heat shock (42˚C, 20 minutes). A: The graph shows the differences in deuteration of individual peptides between heat-shocked HSF1 and control HSF1 monomer. HDX was measured at time intervals of 30, 600 and 3600 seconds. B: The graph shows the percentage of deuteration in HSF1 (y-axis) over the 30-second time interval. Purified HSF1 monomer before (blue line) and after (green line) heat stress (42˚C, 20 minutes), compared to purified HSF1 trimer (red line).

## Macromolecular crowding promotes trimerization of HSF1 and enhances its DNA binding activity

Our hypothesis suggests that HSF1 functions as a direct sensor of proteotoxic stress, with its conformational change and activation triggered by physical conditions leading to unfolded protein accumulation and increased chaperone demand. This phenomenon is particularly

prevalent in metabolically active cells, where HSF1 activation is heightened, but is limited in nutrient-deprived and quiescent cells [14–16]. Additionally, other studies have shown HSF1 inhibition by mTOR signaling or ribosomal inhibitors, as well as AMPK activation [16, 17], highlighting the significance of increased proteosynthesis and unfolded proteins in HSF1 activation. To evaluate the effect of protein crowding on purified HSF1, we used BSA as a natural protein crowder at concentrations relevant for mimicking in vivo crowding effects [18]. Analysis of the impact of both time and crowder concentration on HSF1 trimerization in vitro, conducted using Blue native electrophoresis and western blotting, revealed that low BSA concentrations stabilized HSF1 monomers. However, increasing BSA concentrations facilitated HSF1 trimerization at slightly elevated, near-physiological temperatures (Fig 4A and 4B). This induction of trimerization by elevated BSA concentrations supports the concept of direct HSF1 activation through protein crowding. To support the previous results and to assess whether crowding-induced trimers were properly folded independently of chaperones, we conducted EMSA experiments. Using purified monomeric HSF1 and Cy5-labeled HSE oligonucleotide, we investigated the DNA binding capacity of HSF1 under conditions of mildly elevated temperatures and protein-based crowding environment. These results demonstrate that a crowding environment not only facilitates HSF1 trimerization, as observed by native electrophoresis, but also enhances its DNA binding activity (Fig 4C and 4D; see S3 Fig for additional temperatures).

To explore the effect of crowding in living cells, we subjected cells to high osmotic pressure using sorbitol, a compound that does not penetrate the plasma membrane, leading to cell shrinkage and increased intracellular crowding [19]. Firstly, we confirmed the nuclear localization of HSF1 under normal conditions (S4 Fig). Notably, Fig 4E shows that sorbitol concentrations above 250 mM cause HSF1 to localize into granules within the nucleus. High resolution clear native electrophoresis (HR CNE) further revealed that HSF1 trimers form within 20 minutes of exposure to sorbitol concentrations exceeding 100 mM (Fig 4F). To investigate the dynamics of this process, we exposed H1299 cells expressing mCherry-HSF1 to 0.3 M sorbitol for varying times and analyzed HSF1 oligomerization using HR CNE. The results demonstrate that HSF1 forms trimers after just 10 minutes of sorbitol exposure (S5 Fig). To confirm that these oligomers are DNA-binding competent trimers, we combined EMSA with HR CNE to study trimerization and DNA binding in parallel (S5 Fig). The findings show that HSF1 not only shifts to a higher molecular weight in response to sorbitol but also forms trimers capable of DNA binding. These observations indicate rapid initiation of HSF1 activation following sorbitol treatment, which correlates with cell shrinkage and increased molecular crowding. To further validate the impact of crowding in the cellular environment, we investigated whether reducing crowding could prevent HSF1 activation.

## Inhibition of protein synthesis prevents activation of HSF1

Previous results highlight that protein crowding amplifies the impact of proteotoxic stress on HSF1 trimerization. This phenomenon is further enhanced by increased hydrophobic interactions of unfolded proteins within the crowded environment [20]. To dissect the role of unfolded proteins, we employed translation inhibitors, which reduce the pool of unfolded proteins and alleviate macromolecular crowding [21]. Subsequent experiments investigated whether translation inhibition could prevent HSF1 activation under proteotoxic stresses, including Hsp90 inhibition, proteasome inhibition, and elevated temperature. An important question was whether translation inhibitors prevent HSF1 activation in the absence of Hsp90 protection. We employed diverse inhibitors of translation, including rapamycin (targeting mTOR pathway), rocaglamide (translation initiation) [22], and cycloheximide/anisomycin

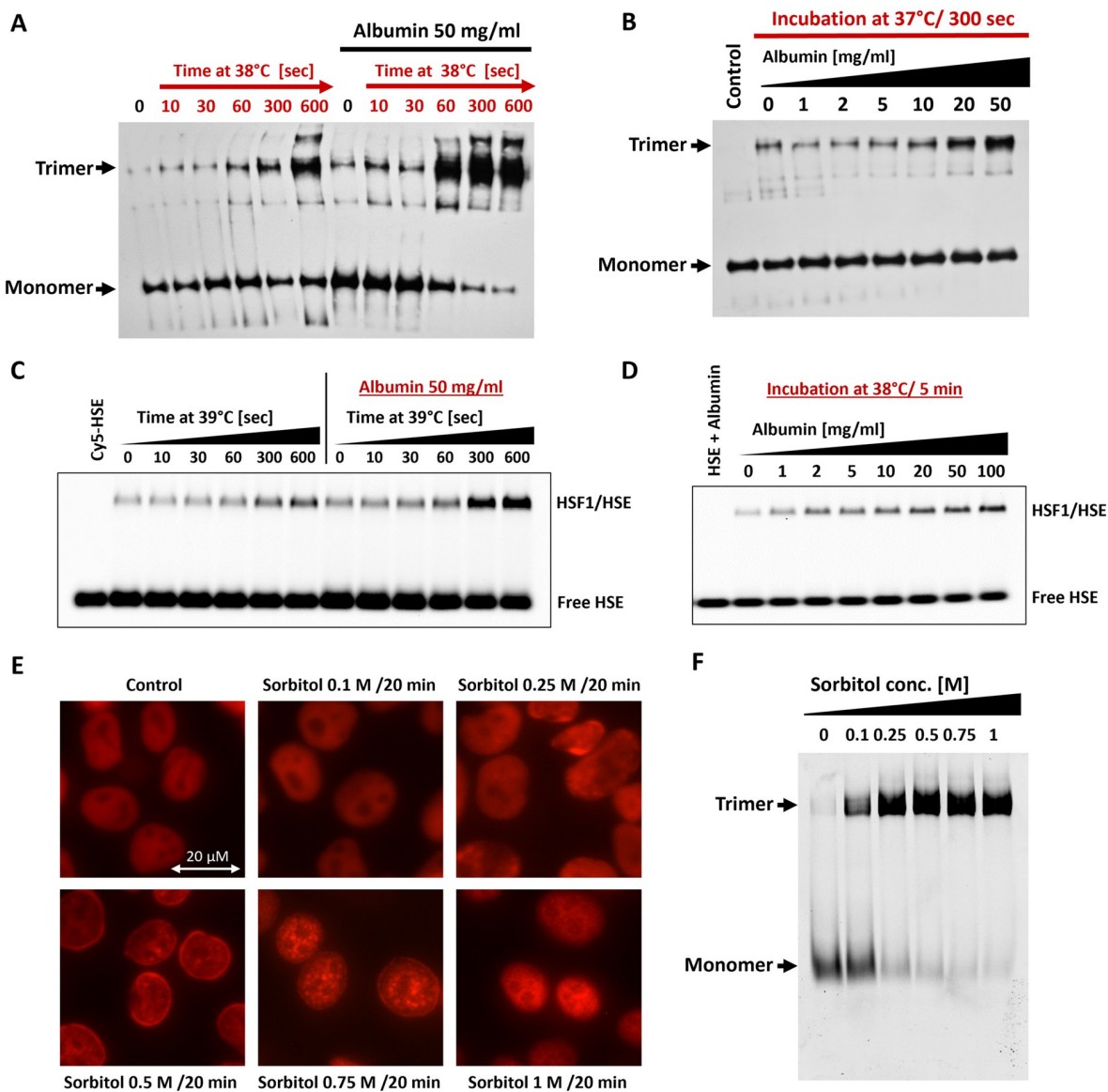

**Fig 4. Excessive crowding leads to HSF1 activation.** A: Native electrophoresis of purified HSF1 monomer (5 μM) after mild heat shock (38°C) in the absence or presence of BSA (50 mg/ml). HSF1, with or without BSA, was incubated at 38°C for increasing times (0–600 seconds). Blue native electrophoresis and western blotting were used to assess HSF1 trimerization. B: Blue native electrophoresis of HSF1 monomer (5 μM) after increasing crowding (BSA 1–50 mg/ml, 5 minutes at 37°C) followed by western blotting. C: EMSA showing HSF1 DNA binding in response to elevated crowding. Monomeric HSF1 was incubated with Cy5-HSE, with or without 50 mg/ml BSA, at 39°C for increasing times. The agarose gel indicates enhanced DNA binding under crowded conditions. D: EMSA of HSF1 monomer mixed with Cy5-HSE, showing the effect of increasing albumin concentrations (0–100 mg/ml) at 38°C for 5 minutes. The gel reveals an increase in DNA binding correlating with higher levels of albumin. The first lane, containing 50 mg/ml BSA mixed with 100 nM Cy5-HSE, serves as a control to demonstrate that BSA alone does not bind to the HSE sequence. E: Fluorescence microscopy showing changes in nuclear localization of HSF1 in H1299 cells stably expressing HSF1 fused to mCherry after 20 minutes exposure to 0.1 to 1M sorbitol in standard culture medium. F: HR CNE native electrophoresis of cell lysates containing mCherry-HSF1 after exposure to sorbitol-supplemented medium. See S1 Raw images for raw data.

(translation elongation) [23]. HR CNE revealed that inhibiting translation effectively halts HSF1 trimerization across all tested stress conditions (Fig 5A and 5F). To mitigate concerns related to cell lysis artifacts, we monitored HSF1 activation in live cells, where, in alignment with previous experiments, inhibition of translation by cycloheximide prevented the formation

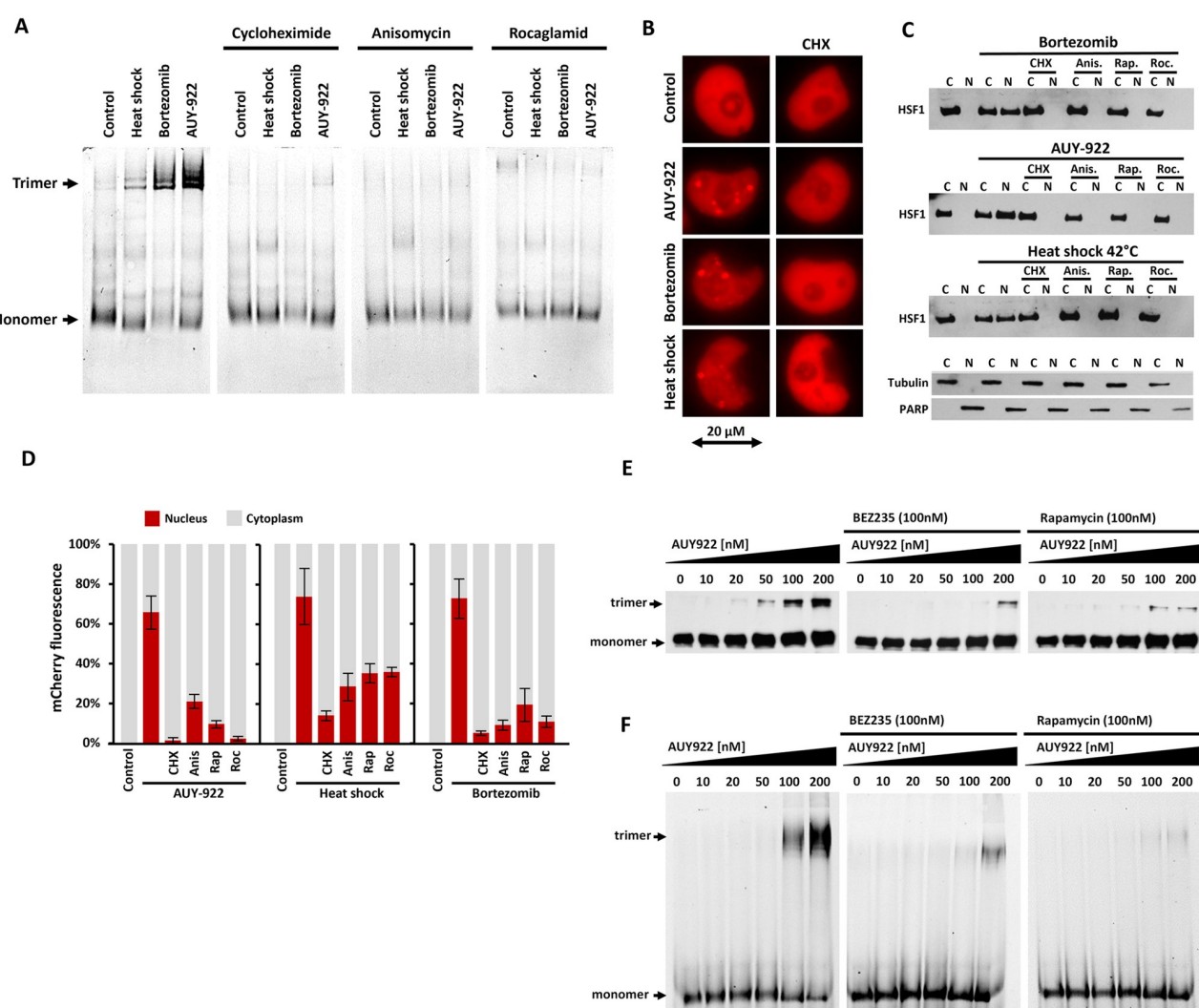

**Fig 5. Effect of proteosynthesis inhibition on HSF1 activation.** A: H1299 cells expressing mCherry-HSF1 were exposed to 42˚C heat shock, the proteasome inhibitor bortezomib (300 nM) and the Hsp90 inhibitor AUY-922 (300 nM) for 1 hour. HR CNE native electrophoresis shows inhibition of trimerization upon simultaneous inhibition of proteosynthesis by cycloheximide (CHX) 10 μg/ml, anisomycin 40 μM and rocaglamide 100 nM. B: HSF1 activation in living cells was determined by the detection of nuclear stress bodies. Nuclear distribution of mCherry-HSF1 was detected after 1 hour of treatment with AUY-922 (300 nM), bortezomib (300 nM), and heat shock at 42˚C, with or without simultaneous application of cycloheximide (CHX). C: The effect of proteosynthesis on HSF1 activation was tested by fractionation of H1299 cells containing endogenous HSF1. Cells were exposed to 42˚C, bortezomib (300 nM), or AUY-922 (300 nM) for 1 hour, with protein synthesis inhibitors cycloheximide (CHX) 10 μg/ml, 40 μM anisomycin (Anis), 10 nM rapamycin (Rap.), or 100 nM rocaglamide (Roc.). Cell lysates were fractionated into nuclear (N) and cytoplasmic (C) fractions. Activated HSF1 bound to DNA is detected in the nuclear fraction, while inactive monomers are present in the cytoplasmic fraction. The level of HSF1 in each fraction was determined by SDS PAGE and detected by blotting using anti-HSF1 mouse mAb c-5 (Santa Cruz Biotechnology Inc.). The efficiency of fractionation was monitored using PARP as a nuclear control and tubulin as a cytoplasmic control. D: Cell fractionation was also used to test the activation of mCherry-HSF1. The graph showing the relative amounts of mCherry-HSF1 in the nuclear (red bar) and cytoplasmic fractions (grey bar) was obtained by measurement of mCherry fluorescence. E: Effect of mTOR inhibitors on HSF1 activation at increasing concentrations of the Hsp90 inhibitor AUY-922. H1299 cells with endogenous HSF1 were pretreated with the mTOR inhibitors BEZ-235 and rapamycin (100 nM) for 8 hours. The effect of AUY-922 was monitored 2 hours after adding it at concentrations of 0–200 nM. HSF1 trimerization was detected by protein crosslinking by adding 2 mM DSBU (disuccinimidyl dibutyric urea) to the lysate for 30 minutes. HSF1 was resolved by SDS-PAGE and detected with mouse monoclonal antibody c-5. F: HSF1-mCherry trimers in identically treated cells were detected by HR CNE. See S1 Raw images for raw data.

of nuclear stress bodies (Fig 5B). To exclude artifacts associated with fluorescent labeling, we conducted cell fractionation experiments using H1299 cells containing endogenous HSF1 subjected to the same conditions (Fig 5C). Cell lysates were fractionated into nuclear (N) and cytoplasmic (C) fractions. Activated HSF1 bound to DNA was detected in the nuclear fraction, while inactive HSF1 was found in the cytoplasmic fraction. Cell fractionation was also used to assess the activation of mCherry-HSF1 (Fig 5D). Fig 5C analyzes endogenous HSF1 in the maternal line, while Fig 5D measures total fluorescence from exogenously introduced mCherry-HSF1, which may result in higher non-specificity due to overexpression of a tagged protein. However, the effect of translation inhibitors on blocking HSF1 activation is evident in both cases. Our results suggest that reducing proteosynthesis modulates HSF1 activation by decreasing molecular crowding. Furthermore, they suggest a mechanism wherein a crowded environment and unfolded proteins destabilize the monomeric conformation of HSF1, prompting its trimerization. Consequently, protein synthesis inhibitors diminish unfolded proteins, thereby reducing crowding and preventing the destabilization of monomeric HSF1.

## Regulation of HSF1 activation independently of chaperones

An intriguing observation was the nullification of HSF1 activation induced by Hsp90 inhibition upon proteosynthesis inhibition, challenging the model in which Hsp90 binding is necessary to maintain its monomeric state. This model proposes that HSF1 remains bound to Hsp90 in its monomeric state, and disruption of this interaction is necessary for HSF1 activation. Consequently, Hsp90 inhibition should activate HSF1 independently of proteosynthesis. To further investigate the role of Hsp90 or Hsp70 in HSF1 activation, we performed streptavidin pull-down experiments utilizing H1299 cells expressing either SBP-tagged Hsp70 or Hsp90. Cells were treated with proteotoxic stressors, with or without the proteosynthesis inhibitor cycloheximide. Subsequently, we isolated SBP-tagged chaperones along with their substrates and analyzed HSF1 levels via western blotting.

Remarkably, chaperone depletion emerged as not the primary factor in activating HSF1. Instead, exposure to heat stress and bortezomib led to increased interaction with Hsp70 (Fig 6A), while the interaction with Hsp90 remained stable (Fig 6B). Furthermore, this interaction persisted after treatment with translation inhibitors, which maintained HSF1 in its monomeric state (Fig 6). The disruption of the HSF1-Hsp90 interaction following Hsp90 inhibition and translation inhibition suggests that Hsp90 is not directly responsible for maintaining HSF1 in its monomeric form. Increased interaction of HSF1 with Hsp70 after stress is consistent with in vitro experiments showing that Hsp70 preferentially binds to trimeric HSF1, subsequently unzipping its trimeric structure [24]. Additionally, Hsp90 is shown to be unable to convert trimeric HSF1 into monomers [4]. Therefore, our cell-based experiments indicate that while chaperones interact with HSF1 and facilitate conformational changes, the final conformation and trimerization are determined by the physical conditions in the cytoplasm. In other words, chaperones help achieve a state of minimal free energy, with the monomeric or trimeric state being determined by external physical conditions such as molecular crowding or temperature.

## Hsp90 inhibition induces cytoplasmic changes, increasing impedance and activating HSF1, which can be blocked by inhibiting proteosynthesis

We hypothesized that HSF1 activation is directly linked to physical changes in the cytosol induced by Hsp90 inhibition. To validate this hypothesis and understand how Hsp90 inhibition triggers HSF1 activation through stress caused by denatured proteins, we developed a method to assess cytoplasmic changes associated with HSF1 trimerization. In previous studies investigating the cytotoxic effect of Hsp90 inhibitors, we observed a pronounced increase in

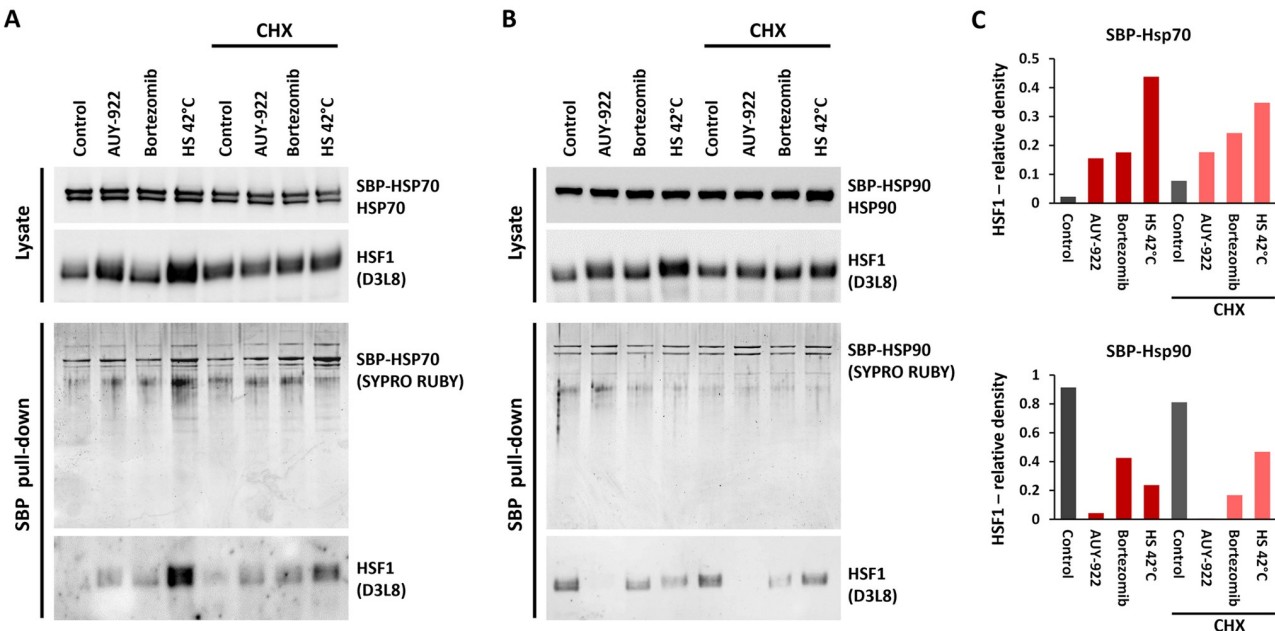

**Fig 6. Interaction of HSF1 with chaperones under stress and translation inhibition.** A: SBP-pull down assay conducted on H1299 cells expressing SBP-Hsp70. Cells were exposed to cycloheximide followed by treatment with AUY-922, bortezomib, or heat shock. The upper panel illustrates levels of SBP-Hsp70 and HSF1 in the cell lysate. The lower panel depicts levels of eluted proteins assessed by SYPRO Ruby staining and HSF1 determined by western blotting. B: SBP-pull down assay analogous to (A), utilizing H1299 cells expressing SBP-Hsp90. The figures illustrate the dynamics between chaperones and HSF1 under various conditions modulating HSF1 activity. See S1 Raw images for raw data. C: Densitometric analysis of HSF1 in complex with Hsp70 and Hsp90. The graph shows the relative amounts of eluted HSF1, normalized to the intensity of HSF1 bands in the lysate.

cell impedance. Utilizing a widely used multi-well format system (xCELLigence), primarily employed for measuring cell proliferation and viability based on impedance changes due to increased cell number, we explored alterations not only in cell count but also in individual cell behavior associated with modified cytoplasmic resistance and capacitance [25]. Additionally, experiments on purified proteins revealed that protein denaturation exposes hydrophobic amino acids to solvent, altering their dielectric properties and resulting in elevated impedance [26]. Our initial experimental data demonstrated that Hsp90 inhibitors elevate impedance despite suppressing cell proliferation, suggesting independence from changes in cell count or morphology (Fig 7). This increase in impedance aligns with the accumulation of unfolded proteins in cells following Hsp90 inhibition. Using various structurally distinct Hsp90 inhibitors (17-AAG, AUY-922, Ganetespib, Olanespib and BIIB21), we investigated if impedance changes were specifically induced by Hsp90 inhibition and not due to nonspecific inhibitor activity (e.g. altered ionic state of cells). All five Hsp90 inhibitors increased cell impedance within the timeframe corresponding to HSF1 activation. To exclude the possibility of extracellular Hsp90 affecting impedance, we employed the cell membrane impermeable Hsp90 inhibitor Geldanamycin-FITC as a negative control (Fig 7A). To demonstrate that proteosynthesis inhibition mitigates stress induced by Hsp90 inhibitors, we measured cell impedance after Hsp90 inhibition while inhibiting proteosynthesis. Protein synthesis inhibitors, including cycloheximide, rocaglamide, or mTOR inhibitors, suppressed the increase in impedance after Hsp90 inhibition (Fig 7B).

To confirm that increased impedance and HSF1 activation result from protein denaturation after Hsp90 inhibition and not HSF1-driven gene expression, we conducted experiments

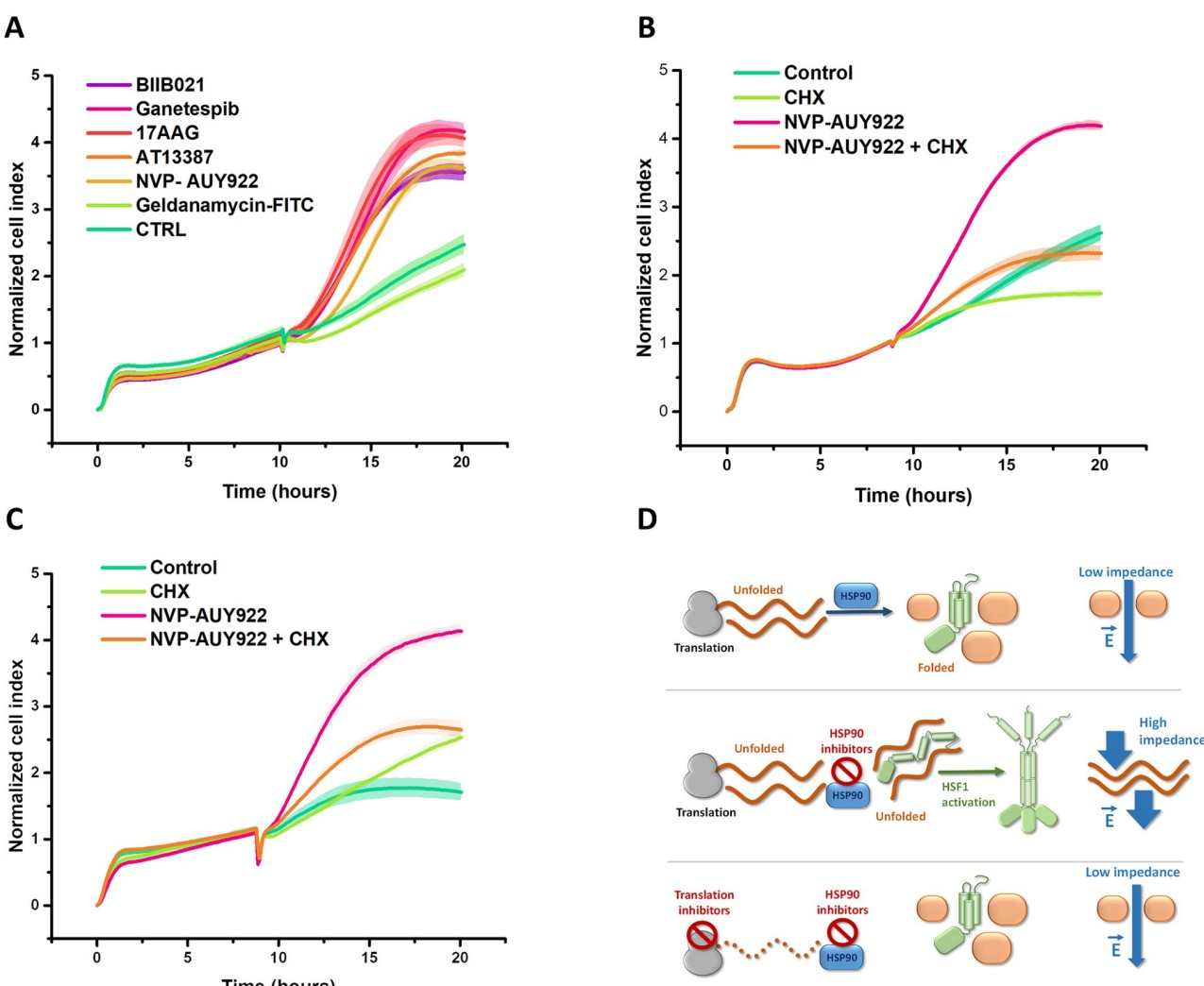

**Fig 7. Impedance in response to Hsp90 inhibitors.** A: Cells were treated 10 hours after seeding with different Hsp90 inhibitors. B: The effect of Hsp90 inhibition and protein synthesis inhibition by cycloheximide (CHX) 10 µg/ml. C: H1299 cells with HSF1 and HSF2 gene knockout. The impedance measurements were performed in four biological replicates, and the standard deviation of the measurements is expressed in the graph by the lighter color filling the area around the curve. D: Experimental setup showing correlation between the presence of denatured proteins after Hsp90 inhibition, impedance, and activation of HSF1.

using HSF1/HSF2 knockout cells. Fig 7C illustrates that Hsp90 inhibition increases cell impedance regardless of HSF1 and HSF2 activity. These results validate that impedance reflects global changes in cell properties following Hsp90 inhibition, attenuated by protein synthesis inhibition (Fig 7D). The impedance-based assessment suggests that cytosolic changes post Hsp90 inhibition directly activate HSF1. While Hsp90 inhibition leads to denatured protein accumulation, impedance increase, and HSF1 activation, blockade of proteosynthesis during Hsp90 inhibition prevents denatured protein accumulation, manifested by impedance decrease and HSF1 activation prevention. These findings support our hypothesis that HSF1 activation upon Hsp90 inhibition arises from physical stress, modulated by proteosynthesis inhibition.

## Discussion

Understanding the cellular responses to proteotoxic stress is crucial for elucidating mechanisms governing cellular homeostasis. The activation of HSF1 in response to diverse stress conditions that cause damage at the protein level indicates its role as a key sensor of proteotoxic stress. However, the precise molecular mechanisms underlying HSF1 activation, especially regarding whether a common factor triggers its activation across diverse stress conditions [27], remain elusive. Our study demonstrates that HSF1 undergoes a conformational change, transitioning from an inactive monomeric form to an active trimeric form, which binds to DNA and initiates transcription in response to stress conditions.

Surprisingly, our in vitro experiments indicate that chaperone activity may not be necessary to maintain the inactive monomeric state of HSF1 under physiological conditions. We observed that stressors such as heat or increased macromolecular crowding can induce HSF1 trimerization in the absence of other proteins and independently of post-translational modifications. These findings suggest that HSF1 acts as a primary sensor of proteotoxic stress, relying on stress-induced conformational changes for activation.

To explore the influence of the crowded cellular environment on HSF1 activation, we first tested its response to elevated levels of BSA, a natural protein crowder. Our results suggest that macromolecular crowding, coupled with mildly elevated temperature, promotes HSF1 activation. The concept of crowding has emerged as a factor promoting protein stabilization since early experiments investigating the effects of synthetic polymers on proteins. This phenomenon has been attributed to the restrictive influence of the repulsion of hard spheres, which limits the range of conformational states available to proteins. Consequently, proteins are more likely to adopt their folded states [28]. Subsequent research has revealed a more intricate impact of crowding on protein conformation, especially when interactions occur between the crowding agents and the studied protein [29–32].

This effect becomes notably pronounced when the proteins themselves represent the crowding agent. In this case, the stabilizing effect of hard-core steric repulsions can be opposed by nonspecific soft attractions between crowders and the protein, favoring the unfolded state energetically [20, 33, 34]. When interpreting the effect of crowding on HSF1, it should be taken into account that its activation does not consist only of trimer assembly but involves the breakdown of the monomeric conformation, which is followed by trimer assembly through the hydrophobic HR-A/B regions. Based on studies exploring the impact of protein crowders on protein conformation, we propose that both soft attractions and hard sphere repulsions might contribute to the conformational switch and trimerization of HSF1.

Furthermore, another factor that potentiates trimerization, along with crowding, is increased temperature. Studies on the effects of temperature and hydrostatic pressure on the dynamical properties and folding stability of highly concentrated lysozyme solutions revealed that elevated temperatures can destabilize protein structure due to transient destabilizing intermolecular interactions. This effect is particularly noticeable under conditions at very high protein concentrations, which mimic densely packed intracellular environments [35].

Surprisingly, our in vivo experiments revealed that HSF1 activation subsequent to Hsp90 inhibition is attenuated by reducing protein synthesis. This outcome is consistent with previous studies showing a tight coordination between protein translation and HSF1 activation [17]. Moreover, this observation challenges the hypothesis that Hsp90 is necessary to maintain the monomeric state of HSF1 [8]. Instead, it raises the possibility that HSF1 undergoes direct activation due to macromolecular crowding and the presence of unfolded proteins.

The interaction studies demonstrated that both Hsp90 and Hsp70 interact with HSF1 following stress exposure. Notably, HSF1's interaction with Hsp70 intensifies under stress,

potentially due to monomeric conformational alterations and hydrophobic domain exposure. Importantly, the degree of interaction between HSF1 and Hsp70 remains consistent after inhibiting translation when stress does not lead to the formation of trimeric HSF1. Although Hsp90 inhibition disrupts the HSF1 complex, translation inhibition still prevents trimer formation. These findings suggest that the conformational state of HSF1 is likely governed by the free energy landscape, shaped by properties of the crowded environment. While chaperones do not appear to dictate the final conformation of HSF1, they may facilitate its transition to a state of minimal free energy (Fig 8). This explanation is supported by experiments conducted on purified HSF1, where the presence of Hsp90 not only failed to hinder trimerization but seemed to facilitate HSF1's transition to trimers under elevated temperatures [4].

When other unfolded proteins surround a protein, nonspecific hydrophobic interactions can occur, preventing its proper folding and leading to aggregation, which is further exaggerated in a crowded intracellular environment [20, 36]. Thus, under crowding conditions in vivo, the presence of unfolded proteins becomes part of the environmental conditions that determine the structural stability and conformational state of proteins. The effect of unfolded proteins on their surroundings was previously described as conformational propagation with prion-like characteristics [37–39].

We propose a hypothesis that unfolding multiplication under crowding may drive conformational changes in monomeric HSF1, leading to trimer assembly. Moreover, protein folding occurs in the context of translation, in which nascent polypeptides are exposed to the cellular environment in an incomplete state, lacking the structural information needed for stable folding [40]. Recent works revealed that the speed of translation can affect the folding efficiency of proteins; slowing down translation improves protein folding, while increasing the elongation rate reduces folding efficiency [41–43].

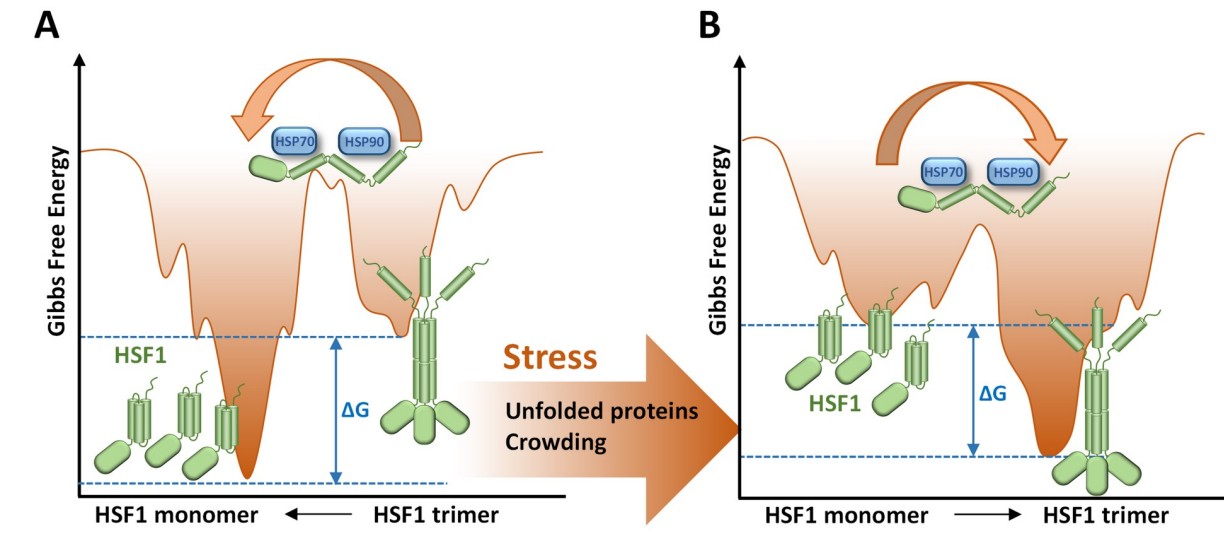

**Fig 8. Altered energetic landscape by crowding induces HSF1 conformational change.** The folding funnel illustrates the energy landscape and maps protein states based on free energies. It assumes the native state has the lowest energy, forming a funnel with the folded protein at the bottom. Landscape shape changes with physical conditions, including unfolded proteins under crowding conditions. A: The proposed model suggests that folding of HSF1 into monomers is more favourable under physiological conditions. B: Increased crowding and unfolded proteins encourage conformational changes, favoring trimeric HSF1. Molecular chaperones like Hsp70 or Hsp90 might aid in surpassing energy barriers to reach the lowest energy state.

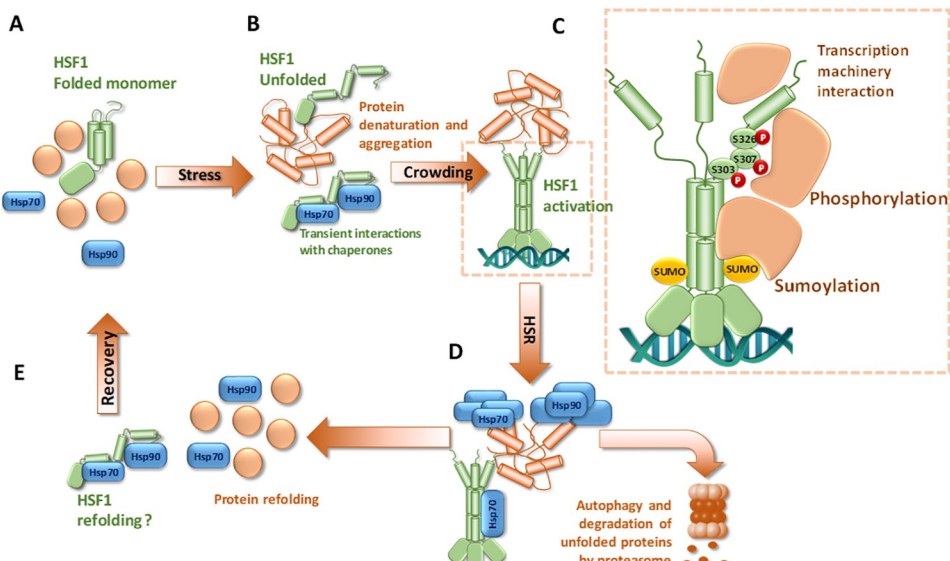

**Fig 9. The effect of stress and molecular chaperones on HSF1 activation.** A: HSF1 is a stable monomer under physiological conditions. B: Proteotoxic stress is associated with increased crowding that results in destabilization of HSF1 monomers. C: Destabilization of monomer structure induces formation of HSF1 trimers that bind DNA and undergo extensive posttranslational modifications. D: Gene expression driven by activated HSF1 restores proteostasis by enhancing refolding and/or degradation of damaged/denatured proteins. E: Restoration of natural conditions enables HSF1 refolding into the stable monomeric conformation.

This observation aligns with the principal role of HSF1 in inducing the synthesis of chaperones that subsequently protect against the negative impacts of misfolded proteins. Chaperones reduce the negative effects of crowding in the presence of misfolded proteins by binding to their exposed hydrophobic regions and preventing their nonspecific interaction and aggregation [44]. High proteosynthesis, resulting in the generation of nascent unfolded proteins, along with stress conditions, may lead to the depletion of chaperone binding capacity [45]. The resulting excess of unfolded proteins in the crowded intracellular environment may trigger a conformational change of HSF1, leading to its activation followed by the expression of protective chaperones. While chaperone activity does not seem to directly influence the equilibrium between monomeric and trimeric HSF1, we assume that chaperones play a role in restoring normal intracellular conditions by facilitating the refolding or degradation of damaged proteins. Thus, in addition to our findings on the direct activation of HSF1 by proteotoxic stresses, the previously proposed negative feedback loop between chaperones and HSF1 remains relevant as part of the complex cellular responses to stress (Fig 9).

## Supporting information

**S1 Fig. Peptide uptake plots for HSF1 conformational states.** A list of peptide uptake plots for individual peptides of HSF1. The data compare deuteration levels over time for HSF1 monomer, HSF1 monomer after 42˚C heat shock, and HSF1 trimer.
(PDF)

**S2 Fig. Analysis of purified HSF1.** A: SEC chromatogram of HSF1 was compared to those of standard proteins, BSA (green) and IgG (blue), each loaded at 1 mg/ml. SEC was conducted under identical conditions for HSF1, BSA, and IgG. B: SEC fractions were analyzed by

SDS-PAGE followed by Coomassie staining. C: Correlation curves from dynamic light scattering (DLS) measurements are presented for HSF1 monomer (fraction C11) and trimer (fraction C2) samples. All measurements were conducted in triplicates at 25˚C. D: Size distribution profiles obtained from DLS for HSF1 monomer (fraction C11) and trimer (fraction C2) samples. The size distribution data shows a median of 11.35 nm with a standard deviation of 2.802 nm for the monomer, and a median of 21.14 nm with a standard deviation of 6.169 nm for the trimer. (TIF)

**S3 Fig. Effect of BSA on HSF1 DNA binding at different temperatures.** A: EMSA analysis showing the effect of increasing concentrations of BSA on HSF1 DNA binding at 4˚C. HSF1 monomer (200 nM) and Cy5-HSE (100 nM) were incubated with BSA (0–100 mg/ml) for 30 minutes at 4˚C. Results indicate that BSA does not impact DNA binding at this temperature. B: Comparison of HSF1 DNA binding with or without 50 mg/ml BSA at 38˚C, across different incubation times (0–600 seconds). The data show that BSA influences HSF1 DNA binding kinetics at mildly elevated temperatures. (TIF)

**S4 Fig. Nuclear localization of HSF1 in H1299 cells under normal conditions.** Microscopy images of H1299 cells expressing mCherry-HSF1. Cells were stained with Hoechst (1 μg/ml) for 15 minutes to visualize nuclei. The figure includes two representative cells A/B. The data confirm that under normal conditions, HSF1 exclusively localizes to the nucleus. (TIF)

**S5 Fig. Effects of sorbitol and proteotoxic stress on HSF1 trimerization and DNA binding.** A: HR CNE showing the formation of HSF1 trimers in H1299 cells expressing mCherry-HSF1 after treatment with 0.3 M sorbitol for varying times (0–30 minutes). Trimer formation is observed after 10 minutes of sorbitol exposure. B: RT-qPCR analysis of Hsp70 expression in H1299 cells exposed to sorbitol. AUY-922 was used as a positive control. Although sorbitol leads to initiation of HSF1 activation, it does not result in the full activation of the heat shock response. C: EMSA employing HR CNE showing the effect of proteotoxic stress on HSF1 trimerization and DNA binding. H1299 cells expressing GFP-HSF1 were treated with 300 nM AUY-922, 300 nM bortezomib and heat shock 42˚C for 1 hour, or 0.3 M sorbitol for 20 minutes. Cell lysates were incubated with 100 nM Cy5-HSE for 30 minutes at 4˚C and analyzed by HR CNE and fluorescence imaging using a Typhoon FLA 9500. The native gel shows HSF1 trimerization under stress conditions, while EMSA confirms the ability of trimers to bind DNA. See S1 Raw images for raw data. (TIF)

**S1 File. pLENTI-N-SBP-TEV-HSP70-IRES-EmGFP-GW.**
(TXT)

**S2 File. pLENTI-N-SBP-TEV-HSP90a-IRES-EmGFP-GW.**
(TXT)

**S3 File. pEXP17-HSF1-wt-GWs.**
(TXT)

**S4 File. pLenti6–3-N-Cherry-Puro-GW-Dest.**
(TXT)

**S5 File. pLenti-N-mCherry-rbs-HSF1-wt-Puro.**
(TXT)

**S6 File. pENTR_TEV_HSF1_GWs.**
(TXT)

**S1 Raw images. Uncropped blots and gel scans underlying all figures presented in the manuscript.**
(PDF)

## Author Contributions

**Conceptualization:** Oliver Simoncik, Petr Muller.

**Formal analysis:** Oliver Simoncik.

**Funding acquisition:** Borivoj Vojtesek, Petr Muller.

**Investigation:** Oliver Simoncik, Vlastimil Tichy, Michal Durech, Lenka Hernychova, Filip Trcka, Lukas Uhrik, Miroslav Bardelcik, Petr Muller.

**Methodology:** Oliver Simoncik, Vlastimil Tichy, Michal Durech, Filip Trcka, Lukas Uhrik, Miroslav Bardelcik.

**Supervision:** Borivoj Vojtesek, Petr Muller.

**Validation:** Oliver Simoncik, Lenka Hernychova.

**Writing – original draft:** Oliver Simoncik, Petr Muller.

**Writing – review & editing:** Philip J. Coates.

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
