## [Decision Letter · Decision Letter 0]

10 Jun 2024

PONE-D-24-16853Direct activation of HSF1 by macromolecular crowding and misfolded proteinsPLOS ONE

Dear Dr. Muller,

Thank you for submitting your manuscript to PLOS ONE. After careful consideration, we feel that it has merit but does not fully meet PLOS ONE’s publication criteria as it currently stands. Therefore, we invite you to submit a revised version of the manuscript that addresses the points raised during the review process.

**Dear Authors,**

**Thank you for submitting your manuscript. Upon review, it is clear that some revisions are necessary to fully support your findings before potential publication. The reviewers have pointed out areas where additional validation could enhance the robustness of your conclusions, particularly concerning the interpretation of specific molecular events. We kindly suggest incorporating further experimental validations, such as molecular weight standards and orthogonal analytical methods. Moreover, it would be beneficial to clarify any discrepancies in the experimental data and ensure that interpretations are well-supported by quantitative evidence. We appreciate your attention to these details and look forward to your revised submission.**

We look forward to receiving your revised manuscript.

Kind regards,

Asif Ali

Academic Editor

PLOS ONE

Journal Requirements:

2. In your Methods section, please report the source of H1299 and A375 human cancer cells used for your study.

The project was supported by the project National Institute for Cancer Research (Programme EXCELES, ID Project No. LX22NPO5102)—funded by the European Union—Next Generation EU and by Ministry of Health Development of Research Organisation, MH CZ - DRO (MMCI, 00209805). O.S., M.B. and P.M. were supported by the Czech Science Foundation (22-17102S), B.V. was supported by the Czech Science Foundation (22-02940S). The funders had no role in study design, data collection and analysis, decision to publish, or preparation of the manuscript.

4. In the online submission form, you indicated that The mass spectrometry proteomics data have been deposited to the ProteomeXchange Consortium via the PRIDE (proteomics identifications database) repository (44) with the dataset identifier “PXDPXD037662”; Username: reviewer_pxd037662@ebi.ac.uk, Password: ChSpggcg.

Further information and requests for resources and reagents should be directed to and will be fulfilled by the Lead Contact, P. Muller (muller@mou.cz).

6. We are unable to open your Supporting Information files "0414-pLENTI-N-SBP-TEV-HSP70-IRES-EmGFP-GW.gbk", "0415-pLENTI-N-SBP-TEV-HSP90a-IRES-EmGFP-GW.gbk", "0617-pEXP17-HSF1-wt-GWs.gbk", "0734-pLenti6-3-N-Cherry-Puro-GW-Dest.gbk", "0756-pLenti-N-mCherry-rbs-HSF1-wt-Puro.gbk", "0812-pENTR_TEV_HSF1_GWs.gbk". Please kindly revise as necessary and re-upload.

Reviewers' comments:

Reviewer's Responses to Questions

**Comments to the Author**

1. Is the manuscript technically sound, and do the data support the conclusions?

Reviewer #1: Partly

Reviewer #2: Partly

2. Has the statistical analysis been performed appropriately and rigorously? 

Reviewer #1: Yes

Reviewer #2: N/A

3. Have the authors made all data underlying the findings in their manuscript fully available?

Reviewer #1: Yes

Reviewer #2: Yes

4. Is the manuscript presented in an intelligible fashion and written in standard English?

Reviewer #1: Yes

Reviewer #2: Yes

5. Review Comments to the Author

**Reviewer #1:** In this manuscript, the authors investigate the mechanisms underlying the activation of HSF1. Using HSF1 purified from bacteria, the authors show that HSF1 can form ‘trimers’ on its own at elevated temperature, which leads to an increase in HSF1’s DNA binding capacity. The authors claim that in vitro molecular crowding, which mimics the intracellular environment during stress conditions, is the key determinant of HSF1 activation and can occur independent of HSP90 titration. Based on these claims, the authors propose that HSP90-mediated regulation of HSF1 is an indirect consequence of the restoration of proteostasis. While the authors are addressing a fundamentally intriguing question and have performed pertinent experiments, I have several concerns regarding the interpretation of the experiments that warrant the claims of the authors.

Major concerns:

1. The authors have overinterpreted their data in several places to the extent that it appears slovenly. For instance,

a. The authors have equated HSF1 trimerization with cellular activation of HSF1 in many places, including the abstract. HSF1 trimerization, while a key step in the process of HSF1-dependent transcriptional activation of its target genes, is not sufficient to drive activation of the genes.

b. How are the authors certain that the higher molecular weight species in Figs. 2A, 2B, 4A, 4B, 4D and 5 are indeed HSF1 trimers? It is known that HSF1 can form oligomers and aggregates. Therefore, it is unclear if the larger molecular species are HSF1 “trimers”, especially when appropriate molecular weight standards are not used or included in the experiments.

c. The authors inhibited protein synthesis and observed a reduction in HSF1 activation, attributing it to the lack of molecular crowding. However, many previous studies have shown that activation of the heat shock response is dependent on the newly synthesized proteins, as they are the key clients of the molecular chaperones. The authors may want to discuss those models as well.

2. In Fig. 2D, how is the concentration of 'trimer' measured with respect to the monomer? In theory, a molar equivalent of a trimer should have 3X more HSF1 molecules than that of a monomer, leading to a much higher protein concentration. The authors must clarify how the concentrations were compared.

Similarly, heating at 38 ˚C for 10 minutes can lead to oligomerization of HSF1, as seen in Fig. 4A. These oligomers may artifactually alter the tumbling rate of the fluorochrome labeled HSE, resulting in enhanced polarization. The authors may want to incorporate an orthogonal approach such as EMSA, which will also reveal the relative sizes of the free and protein-bound DNA.

3. In Fig. 4, as the authors used BSA in conjunction with a higher temperature, which might cause HSF1 to oligomerize on its own, the experiment assessing the effects of BSA in Figure 4 is inconclusive. The authors should test the role of BSA at the monomeric temperature of HSF1 used in Fig. 2C.

4. In Fig. 4C, the osmotic stress the authors use in their molecular crowding experiments causes a cascade of previously identified cell signaling pathways to be activated. Thus, HSF1 may be activated by mechanisms other than molecular crowding in osmotic stress conditions. The authors must use caution in interpreting their data.

Did the authors observe any transcriptional changes in the HSF1 target genes upon inducing osmotic shock? Also, what are the granules in nucleus? HSF1’s localization into the granules may not necessarily mean that HSF1 is active.

5. In Fig. 5C, the authors should include blots of control proteins of the cytoplasmic fraction (e.g., GAPDH) and the nuclear fraction (e.g., histone)

6. The authors claim that their experiments with HSP90 inhibitors in conjugation with translation inhibition calls the chaperone titration model into question. However, as alluded above, the newly synthesized proteins are indeed the key chaperone clients. Also, the HSP90 inhibitor used by the authors not only binds in the ATP pocket of HSP90 but may also alter other cell signaling axes. Combining all of it, the authors have a redundant, non-specific inhibition system that may not necessarily inform the HSP90 mode of action. Indeed, the fact that cytoplasmic restoration of proteostasis contributes to the regulation of HSF1 is far more consistent with their own findings. As is, this reviewer is still not convinced that the data presented by the authors challenge the chaperone titration model.

7. Experiments in Fig. 6 and the accompanying text requires a thorough reevaluation. In Fig. 6A, how do authors reconcile the increase in the association of HSF1 and HSP70 following heat shock and the opposite for HSP90? The levels of HSF1 are widely varying in the lysate, which makes quantitative interpretation of the data very challenging.

8. The experiments in Fig. 7 demonstrate the effect of various HSP90 inhibitors on the intracellular environment, but they provide little insight into how HSF1 is activated. Panels A and B only demonstrate that HSP90 inhibitors increase the amount of misfolded proteins contingent upon protein synthesis. Presence or absence of HSF1 (panel C) seems irrelevant here as the activation of HSF1 occurs downstream of protein misfolding.

Minor concerns:

1. Line 43: The authors discuss on three possible modes of HSF1 activation. The authors may want to include a statement that these three modes are not mutually exclusive as reported in the literature.

2. It will be helpful for the readers if the authors include the sequence of HSE that was used in their fluorescence polarization assays, as well as the equation used to calculate the dissociation constants.

3. In Fig. 3, the authors did not provide any rationale for using 42 ˚C for 20 mins, whereas in most of their other experiments they used ~38 ˚C for 10 mins.

4. Are these western blots in Figs. 4A and 4B? If not, then how are the authors able to differentiate between albumin and HSF1? An explanation of the methodology will be helpful to the readers.

**Reviewer #2: **This study offers intriguing insights into the mechanisms by which the stress responder transcription factor HSF1 is activated. The authors conducted a series of in vitro and in vivo experiments to demonstrate that HSF1 activation is driven by protein crowding which leads to the stress induced conformational changes in HSF1 rather than the previously accepted mechanisms involving chaperone binding or post-translational modifications.

The experiments are well-designed, however, lack proper controls. The following issues need to be addressed.

Figure 2 A, B: The authors show that the peaks in gel filtration correspond to trimeric and monomeric forms of HSF1.To validate that the peaks indeed correspond to trimer and monomer, it is essential to run molecular weight markers alongside the samples. By plotting a standard curve of known molecular weights versus elution volumes, the authors can more accurately determine the molecular weights of the observed peaks. Complementary techniques such as Dynamic Light Scattering (DLS) or Analytical Ultracentrifugation (AUC) should be employed to further substantiate the oligomeric states of HSF1.

Figure 3: The authors suggest a conformational change in HSF1. To provide stronger evidence of the conformational change in HSF1, the authors should perform Circular Dichroism (CD) spectroscopy, which can detect changes in secondary structure. Additionally, fluorescence spectroscopy, particularly using intrinsic tryptophan fluorescence or extrinsic probes such as ANS (8-Anilino-1-naphthalenesulfonic acid), could be used to monitor tertiary structure changes.

Figure 4 A, B: The trimerization of HSF1 in the presence of 50 mg/mL BSA at 38ºC (Fig 4A) and at 37ºC (Fig 4B) shows different extents of trimerization, with a higher amount of monomer present at 37ºC. The discrepancy between the trimerization extents at 37ºC and 38ºC needs to be addressed. The authors should perform additional replicates and statistical analyses to determine if the observed differences are significant and reproducible. Detailed information about the experimental conditions, such as precise temperature control, sample preparation, and timing, should be provided to rule out technical variability. Exploring the effects of incremental temperature changes on trimerization could also be informative. Discussing possible biochemical or biophysical reasons for the observed anomaly in trimerization extent would be valuable. This could include differences in protein stability, aggregation propensity, or interactions with BSA at slightly different temperatures.

Figure 4A. The trimerization of purified Hsf1 increases with time till 600 sec, whereas monomeric Hsf1 decreased with time only in presence of BSA. The author should explain the discrepancy of the observation.

Figure 4 C, D: The results suggest that macromolecular crowding and misfolded proteins activate HSF1, and the interpretation that HSF1 activation is independent of chaperone interaction needs further validation. Supplement the qualitative observations with quantitative data showing the extent of HSF1 activation under various crowding conditions. The choice of a 20-minute duration for observing crowding effects should be justified. It would be beneficial to explore the time-dependence of HSF1 activation by conducting a time-course experiment, assessing activation at multiple time points to understand the kinetics of the response. To robustly conclude that HSF1 activation is independent of chaperone interaction, additional experiments should be performed.

Figures 4C and Fig 5B: The authors should show the nuclear localization of mCherry-HSF1 with nuclear staining dye.

Figure 5C. The western blot data is missing controls. A marker protein for cytosol and nucleus should be included in the blot. A significant amount of Hsf1 is found in the nucleus (Fig 5D) in presence of drug upon heat shock which is not found in Fig. 5C. Also, the distribution of Hsf1 in cytosol and nucleus in control and stress condition is not proportional.

Figure 6 A, B. The western blots of the pull-down assay lack molecular weight markers. The loading control is missing in cell lysate. What is the negative control for this experiment? Only bead without overexpressing Hsp70/Hsp90 cell lysate is essential in the western blot data. The full blot of immunoprecipitated Hsf1 is required to assess the monomeric or trimeric form of Hsf1 bound to chaperone.

According to the chaperone hypothesis binding of chaperones with Hsf1 will be reduced upon application of stress. However, immunoprecipitation of Hsp70/Hsp90 with Hsf1 suggests no alteration of chaperone binding. In contrast, Hsp70 binding to monomeric Hsf1 is enhanced upon heat stress. The western blot data showed induction of Hsf1 upon heat stress although no chaperone induction was found. Have the authors tried to see the binding of trimeric Hsf1 with chaperones in normal and stress conditions. Moreover, what will happen if purified Hsp70 is incubated with purified Hsf1 and see the transition of monomeric to trimeric conformation of Hsf1 upon stress.

6. PLOS authors have the option to publish the peer review history of their article (what does this mean?). If published, this will include your full peer review and any attached files.

Reviewer #1: No

Reviewer #2: No

---

## [Author Response · Author response to Decision Letter 0]

15 Aug 2024

Dear Reviewers, 

We sincerely thank you for your thorough review and valuable feedback on our manuscript. Your insightful comments have been instrumental in improving the quality and clarity of our work. Below, we have addressed each of your comments in detail, outlining the changes made to the manuscript and providing additional explanations where necessary. 

We appreciate the opportunity to enhance our research through your constructive critique and look forward to your further evaluation. 

R1:

Reviewer #1: In this manuscript, the authors investigate the mechanisms underlying the activation of HSF1. Using HSF1 purified from bacteria, the authors show that HSF1 can form ‘trimers’ on its own at elevated temperature, which leads to an increase in HSF1’s DNA binding capacity. The authors claim that in vitro molecular crowding, which mimics the intracellular environment during stress conditions, is the key determinant of HSF1 activation and can occur independent of HSP90 titration. Based on these claims, the authors propose that HSP90-mediated regulation of HSF1 is an indirect consequence of the restoration of proteostasis. While the authors are addressing a fundamentally intriguing question and have performed pertinent experiments, I have several concerns regarding the interpretation of the experiments that warrant the claims of the authors. 

Major concerns: 

1. The authors have overinterpreted their data in several places to the extent that it appears slovenly. For instance, 

a. The authors have equated HSF1 trimerization with cellular activation of HSF1 in many places, including the abstract. HSF1 trimerization, while a key step in the process of HSF1-dependent transcriptional activation of its target genes, is not sufficient to drive activation of the genes. 

b. How are the authors certain that the higher molecular weight species in Figs. 2A, 2B, 4A, 4B, 4D and 5 are indeed HSF1 trimers? It is known that HSF1 can form oligomers and aggregates. Therefore, it is unclear if the larger molecular species are HSF1 “trimers”, especially when appropriate molecular weight standards are not used or included in the experiments. 

c. The authors inhibited protein synthesis and observed a reduction in HSF1 activation, attributing it to the lack of molecular crowding. However, many previous studies have shown that activation of the heat shock response is dependent on the newly synthesized proteins, as they are the key clients of the molecular chaperones. The authors may want to discuss those models as well. 

A: 

Thank you for your thorough and constructive comments on our manuscript. We have carefully considered your feedback and made significant revisions to address your concerns. 

We acknowledge that HSF1 trimerization, while a key step in the process of HSF1-dependent transcriptional activation, is not solely sufficient to drive the activation of target genes. To mitigate any over-interpretation of our data, we have revised the manuscript, including the abstract, to more accurately reflect this understanding. Additionally, we have conducted further experiments to verify the purity and functionality of HSF1 trimers. Specifically, we utilized electrophoretic mobility shift assays (EMSA) with fluorescent-labeled HSE oligonucleotides to confirm DNA binding ability. 

We understand the importance of accurately identifying HSF1 trimers and distinguishing them from oligomers and aggregates. To address this, we performed additional EMSA experiments using Cy5-labeled oligonucleotides with cell lysates. These results, now included in the supporting information files (S4 Fig), demonstrate that the putative trimer can bind to DNA, unlike the monomeric form, thereby excluding the possibility of inactive aggregates. In native gels, we did not use molecular markers because the rate of migration in native electrophoresis with the setup used is affected by charge, conformation, and interaction with the gel phase, in addition to the size of the protein. For this reason, we chose an alternative using fluorescence-labeled oligonucleotides to verify the trimeric DNA binding form of HSF1. 

Additionally, to confirm the transactivation capacity of HSF1 under stress conditions, we performed RT-qPCR to assess induction of Hsp70 (HSPA1). Hsp70 mRNA levels correlate well with HSF1 activation, providing further validation of our findings (S4 Fig). 

We have revised the manuscript to discuss existing models, including the chaperone titration model, which highlights the role of newly synthesized proteins as key clients of molecular chaperones. 

R1: 

2. In Fig. 2D, how is the concentration of 'trimer' measured with respect to the monomer? In theory, a molar equivalent of a trimer should have 3X more HSF1 molecules than that of a monomer, leading to a much higher protein concentration. The authors must clarify how the concentrations were compared. 

A:

Thank you for your insightful comment. The calculation was adjusted to account for the molar amount of both the trimer and the monomer. Specifically, at the same molar concentration, the trimer contains three times the amount of protein compared to the monomer due to its composition of three HSF1 molecules. The trimer fraction was purified using size exclusion chromatography (SEC), ensuring that it exclusively contains the trimeric form without any aggregates or monomer contamination. Similarly, we utilized a fraction of pure monomer. For the monomer sample subjected to heat shock, we assumed that most of the protein would transition to the trimeric form. This assumption is supported by the dissociation constant, which approaches that of the pure trimer. To ensure the quality of the protein, we performed additional experiments, including repeats of SEC, and verified the fractions of the purified protein by dynamic light scattering (DLS). The functional activity of HSF1 was confirmed by electrophoretic mobility shift assay (EMSA) with heat shock elements (HSE), demonstrating that binding in the monomeric fractions to HSE was only detected after heat shock. We have also added a methodological section concerning fluorescence polarization to the manuscript. This section presents the methodology for calculating the dissociation constant and clarifies that we calculated the molar concentration of the whole trimers. 

R1: 

Similarly, heating at 38 ˚C for 10 minutes can lead to oligomerization of HSF1, as seen in Fig. 4A. These oligomers may artifactually alter the tumbling rate of the fluorochrome labeled HSE, resulting in enhanced polarization. The authors may want to incorporate an orthogonal approach such as EMSA, which will also reveal the relative sizes of the free and protein-bound DNA. 

A:

Thank you for your valuable feedback regarding the potential effects of oligomerization on the tumbling rate of the fluorochrome-labeled HSE and the suggestion to incorporate an orthogonal approach such as EMSA. 

In response, we have supplemented the manuscript with additional experiments using electrophoretic mobility shift assay (EMSA). Specifically, we conducted an experiment in a similar setup to the fluorescence polarization assay using Cy-5 labeled HSE oligonucleotide. Furthermore, EMSA was employed as an independent method to observe the effect of albumin on HSF1 trimerization. Unlike native electrophoresis, this experiment demonstrates that trimer formation occurs and is capable of binding to DNA. 

These additional experiments provide a comprehensive validation of our findings, addressing the concerns raised about potential artifacts in the fluorescence polarization measurements and confirming the DNA-binding capability of the HSF1 trimers. 

R1:

3. In Fig. 4, as the authors used BSA in conjunction with a higher temperature, which might cause HSF1 to oligomerize on its own, the experiment assessing the effects of BSA in Figure 4 is inconclusive. The authors should test the role of BSA at the monomeric temperature of HSF1 used in Fig. 2C. 

A:

Thank you for your insightful feedback regarding the potential influence of BSA and temperature on HSF1 oligomerization in Figure 4. 

Our experiments indicate that the conversion rate of HSF1 from monomeric to trimeric forms is influenced by both temperature and albumin concentration. The experimental conditions were carefully chosen to minimize the formation of HSF1 aggregates or coaggregates with albumin. To address the specific concern about the role of BSA at a lower temperature, similar to that used in Fig. 2C, we have conducted additional experiments. 

To confirm that high concentrations of albumin promote the formation of active HSF1 trimers, we utilized an alternative method, electrophoretic mobility shift assay (EMSA). We have supplemented the manuscript with these new experiments, demonstrating that albumin, as a natural crowder, facilitates the formation of HSF1 trimers capable of binding to DNA across a range of temperatures (37-40°C). This is supported by both native electrophoresis and EMSA. 

These additional experiments provide further clarity on the role of albumin and temperature in HSF1 oligomerization and substantiate our findings under varying conditions. 

R1:

4. In Fig. 4C, the osmotic stress the authors use in their molecular crowding experiments causes a cascade of previously identified cell signaling pathways to be activated. Thus, HSF1 may be activated by mechanisms other than molecular crowding in osmotic stress conditions. The authors must use caution in interpreting their data. 

A:

We acknowledge that increased osmotic pressure and cell shrinkage can indeed lead to various cellular changes, including HSF1 activation. However, the primary aim of our experiment was to refute the hypothesis that increased molecular crowding does not lead to HSF1 oligomerization. If our data had shown that increased crowding and cell shrinkage due to osmotic stress did not lead to HSF1 activation, it would have indicated that increased protein crowding alone does not directly activate HSF1. Our findings demonstrate that the results of the osmotic stress experiments align with those of the isolated protein experiments. Therefore, we conclude that excessive crowding is the simplest explanation of HSF1 activation. We have revised the manuscript to reflect this conclusion and included a discussion of possible alternative interpretations of our observations. 

R1:

Did the authors observe any transcriptional changes in the HSF1 target genes upon inducing osmotic shock? Also, what are the granules in nucleus? HSF1’s localization into the granules may not necessarily mean that HSF1 is active. 

A: 

We are aware that trimerization does not necessarily imply activation of HSF1 to transcribe downstream genes. To address the activation of HSF1 following sorbitol treatment, we performed several additional experiments. 

Firstly, we added Cy5-labeled HSE oligonucleotide to native electrophoresis to simultaneously detect trimerization and the ability of HSF1 to bind to DNA. This experiment was conducted in control cells and cells exposed to various stress conditions. The results are presented as supporting information files (S4 Fig) and demonstrate induction of HSF1 DNA binding activity in response to sorbitol as well as other stressors. 

Additionally, we performed quantification of Hsp70 (HSPA1A), which correlates with HSF1 activity. In contrast to Hsp90 inhibition (AUY-922), osmotic stress did not increase Hsp70 expression (S4 Fig). This result implicates the side effects of osmotic stress that are known to inhibit transcription. These effects also cause the aberrant formation of HSF1 granules, which are distinct from classical nuclear stress bodies formed after exposure to other stress conditions. 

We have included the following sentence in the revised manuscript to clarify these findings: "Although osmotic stress-induced increased crowding leads to trimerization of HSF1 and its binding to DNA, the side effects of these conditions prevent efficient transcription initiation." 

R1: 

5. In Fig. 5C, the authors should include blots of control proteins of the cytoplasmic fraction (e.g., GAPDH) and the nuclear fraction (e.g., histone) 

A: 

We appreciate your suggestion. We used Tubulin as the control protein for the cytoplasmic fraction and PARP for the nuclear fraction. The results from these western blot analyses have been added to Fig. 5C. 

R1: 

6. The authors claim that their experiments with HSP90 inhibitors in conjugation with translation inhibition calls the chaperone titration model into question. However, as alluded above, the newly synthesized proteins are indeed the key chaperone clients. Also, the HSP90 inhibitor used by the authors not only binds in the ATP pocket of HSP90 but may also alter other cell signaling axes. Combining all of it, the authors have a redundant, non-specific inhibition system that may not necessarily inform the HSP90 mode of action. Indeed, the fact that cytoplasmic restoration of proteostasis contributes to the regulation of HSF1 is far more consistent with their own findings. As is, this reviewer is still not convinced that the data presented by the authors challenge the chaperone titration model. 

A: 

We appreciate the reviewer's insights. We agree that the chaperone titration model is generally valid in demonstrating that depletion of chaperone capacity leads to physical changes in the cytoplasm, which subsequently activates HSF1 directly. Our paper presents a novel mechanism that differs from the existing chaperone titration model. In our mechanism, chaperones do not directly inhibit HSF1 activation but are essential for maintaining the favorable physical conditions necessary for monomer existence during active proteosynthesis. This is not intended to refute the chaperone titration model but rather to clarify the mechanism through which the negative feedback between HSF1 and chaperones operates.

To enhance readability, we clarify that the chaperone titration model complements the direct activation of HSF1. Specifically, we detail a negative feedback loop between HSF1 and chaperones: under proteotoxic stress, direct activation of HSF1 induces chaperones, which then aid in the degradation or refolding of damaged proteins in a crowded cellular environment. Only when protein homeostasis is re-established can the cytoplasmic conditions allow HSF1 to refold into its monomeric form. 

We also acknowledge the concern regarding the specificity of HSP90 inhibitors. To address this, we used submicromolar concentrations of HSP90 inhibitors that are highly selective for HSP90. We assessed the effect on HSF1 at short intervals, up to 2 hours, to minimize off-target effects. Additionally, we used different structurally unrelated inhibitors of HSP90 (17AAG, AUY922, Radicicol) and observed consistent results across these treatments. This approach ensures that our findings are robust and specifically related to the inhibition of HSP90, supporting our proposed mechanism. 

R1: 

7. Experiments in Fig. 6 and the accompanying text requires a thorough reevaluation. In Fig. 6A, how do authors reconcile the increase in the association of HSF1 and HSP70 following heat shock and the opposite for HSP90? The levels of HSF1 are widely varying in the lysate, which makes quantitative interpretation of the data very challenging. 

A: 

Thank you for your detailed comments on our experiments in Figure 6. 

We are aware that when HSF1 is activated by stress, it can simultaneously be stabilized, leading to varying levels of HSF1 in the lysate. This effect may be further enhanced by post-translational modifications that occur following stress, which can also cause a shift in the HSF1 band. This phenomenon is observed in Figure 6 when stress is not combined with the translation inhibitor cycloheximide. With cycloheximide application, HSF1 levels remain balanced because trimerization does not occur under these conditions. 

The resulting experiment provides honest data from live cells, where the change in HSF1 levels cannot be optimized by different loading, as this would also change the levels of SBP-Hsp70 or SBP-Hsp90, which are used as bait in this experiment. Thus, the most appropriate way to quantify the interactions between Hsp70, Hsp90, and HSF1 is to m

---

## [Decision Letter · Decision Letter 1]

9 Oct 2024

Direct activation of HSF1 by macromolecular crowding and misfolded proteins

PONE-D-24-16853R1

Dear Dr. Muller,

We’re pleased to inform you that your manuscript has been judged scientifically suitable for publication and will be formally accepted for publication once it meets all outstanding technical requirements and answers the reviewer 1 concern to change the tone of discussion and over interpretation.

Kind regards,

Asif Ali

Academic Editor

PLOS ONE

Additional Editor Comments (optional):

Reviewers' comments:

Reviewer's Responses to Questions

**Comments to the Author**

1. If the authors have adequately addressed your comments raised in a previous round of review and you feel that this manuscript is now acceptable for publication, you may indicate that here to bypass the “Comments to the Author” section, enter your conflict of interest statement in the “Confidential to Editor” section, and submit your "Accept" recommendation.

Reviewer #1: (No Response)

Reviewer #2: All comments have been addressed

2. Is the manuscript technically sound, and do the data support the conclusions?

Reviewer #1: Partly

Reviewer #2: Yes

3. Has the statistical analysis been performed appropriately and rigorously? 

Reviewer #1: Yes

Reviewer #2: N/A

4. Have the authors made all data underlying the findings in their manuscript fully available?

Reviewer #1: Yes

Reviewer #2: (No Response)

5. Is the manuscript presented in an intelligible fashion and written in standard English?

Reviewer #1: Yes

Reviewer #2: Yes

6. Review Comments to the Author

Reviewer #1: The authors have partially addressed the concerns about the DNA binding capacity and the identity of HSF1 trimers by using EMSA; however, the authors are advised to address the following concerns:

Their RT-qPCR data shows that the sorbitol treatment does not lead to heat shock gene activation, which is consistent with my previous concern of authors misinterpretating HSF1 trimerization as proxy for HSF1 activation. The authors must use a cautionary tone for such interpretation.

Their is little to no evidence in the manuscript that supports author's claim about increased interaction of HSF1 with Hsp70 after stress. They state that "the increased interaction of HSF1 with Hsp70 after stress is consistent with in vitro experiments showing that Hsp70 preferentially binds to the trimeric HSF1 subsequently unzipping its trimeric structure.” To at least corroborate this assertion, the authors must incorporate appropriate citations . Even for their pull down data from the cells, the authors are basing their assertions on a single replicate which even shows significant variation among the lysates.

The authors continue to fail to demonstrate that molecular crowding by BSA alone is sufficient to cause HSF1 to form trimers. In all of their experiments, the BSA treatments are coupled with high temperature. The absence of gene activation upon sorbitol treatment and in their other data, it is evident that molecular crowding is NOT the simplest explanation for HSF1 activation.

Reviewer #2: (No Response)

7. PLOS authors have the option to publish the peer review history of their article (what does this mean?). If published, this will include your full peer review and any attached files.

Reviewer #1: No

Reviewer #2: No

---

## [Editor Report · Acceptance letter]

24 Oct 2024

PONE-D-24-16853R1 

PLOS ONE

Dear Dr. Muller, 

I'm pleased to inform you that your manuscript has been deemed suitable for publication in PLOS ONE. Congratulations! Your manuscript is now being handed over to our production team.

Kind regards, 

on behalf of

Dr. Asif Ali 

Academic Editor

PLOS ONE